

# A water risk index for portfolio exposure to climatic extremes: conceptualization and an application to the mining industry

Luc Bonnafous[1], Upmanu Lall[1,2], Jason Siegel[2]

[1]Columbia Water Center, Columbia University, New York, 10031, USA
[2]Earth and Environmental Engineering Department, Columbia University, New York, 10031, USA

*Correspondence to*: Luc Bonnafous (lmb2216@columbia.edu)

**Abstract.** Corporations, industries and non-governmental organizations have become increasingly concerned with growing water risks in many parts of the world. Most of the focus has been on water scarcity and competition for the resource between agriculture, urban users, ecology and industry. However, water risks are multidimensional. Water related hazards include
flooding due to extreme rainfall, persistent drought, and pollution, either due to industrial operations themselves, or to the failure of infrastructure. Most companies have risk management plans at each operational location to address these risks to a certain design level. The residual risk may or may not be managed, and is typically not quantified at a portfolio scale, i.e., across many sites. Given that climate is the driver of many of these extreme events, and there is evidence of quasi-periodic climate regimes at inter-annual and decadal time scales, it is possible that a portfolio is subject to persistent, multi-year
exceedances of the design level. In other words, for a multi-national corporation, it is possible that there is correlation in the climate induced portfolio water risk across its operational sites as multiple sites may experience a hazard beyond the design level in a given year. Therefore, from an investor's perspective, a need exists for a water risk index that allows an exploration of the possible space and/or time clustering in exposure across many sites contained in a portfolio. This paper represents a first attempt to develop an index for financial exposure of a geographically diversified, global portfolio to the time-varying risk of
climatic extremes using long daily global rainfall data sets derived from climate re-analysis models. Focusing on extreme daily rainfall amounts and using examples from major mining companies, we illustrate how the index can be developed. We discuss how companies can use it to explore their corporate exposure, and what they may need to disclose to investors and regulators to promote transparency as to risk exposure and mitigation efforts. For the examples of mining companies provided, we note that the actual exposure is substantially higher than would be expected in the absence of space and time correlation of risk as is tacitly assumed usually. We also find evidence for the increasing exposure to climate induced risk, and for decadal variability
in exposure. The relative vulnerability of different portfolios to multiple extreme events in a given year is also demonstrated.

Keywords: extreme rainfall, spatio-temporal correlation of risk, portfolio exposure, mining.



## 1 Introduction

Long term investors, such as Sovereign Wealth Funds, need to account for risks that may manifest themselves over several decades, and hence they may have a very different perspective on risk than short term investors. In particular, they have a growing interest in understanding how climate and environmental risks may impact the companies comprising their investment
portfolios. Scientific projections that climate change may increase the frequency and intensity of extreme rainfall and droughts amplify such concerns. Water-related risks dominate the pathways of exposure to climate variability and change. Consequently, many studies are being commissioned to "downscale" climate change projections to the level of cities or even individual assets as part of an environmental risk analysis. In the process, metrics and pathways of climate and water risk exposure at the asset level are re-assessed, including, in some cases, past exposure and outcomes.

However, site-specific data is often limited and regional climate may exhibit significant quasi-periodic or cyclical variability, with periods ranging from inter-annual (e.g. 3 to 7 years in the case of the El Niño Southern Oscillation) to multi-decadal (e.g. 16-20 years for the Pacific Decadal Oscillation and 40-70 years for the Atlantic Multi-decadal Oscillation cf. (Frankcombe, Von Der Heydt, & Dijkstra, 2010) (McCabe & Palecki, 2006) (Biondi, 2001) (Knight, Folland, & Scaife, 2006) (Gershunov & Barnett, 1998) (Grimm & Tedeschi, 2009) (Nicholson & Kim, 1997) (Cayan, Redmond, & Riddle, 1999) (Risbey, Pook,
McIntosh, Wheeler, & Hendon, 2009) (Verdon, Wyatt, Kiem, & Franks, 2004) (Kiem, Franks, & Kuczera, 2003)). Over the last century, facilities designed to deal with floods and droughts or to control pollution, as well as financial risk instruments such as insurance, have typically been designed with less than 30 years of at-site data. As illustrated in Fig. 1 and quantitatively demonstrated in (Jain & Lall, 2001), if the climate cycle shifts, an estimate of a 100-year event based on a specific 30 years of data may correspond to either a more frequent (e.g., 10-year) or rarer (e.g., 1000-year) event at the site in the succeeding 30
years when that instrument is actually used.





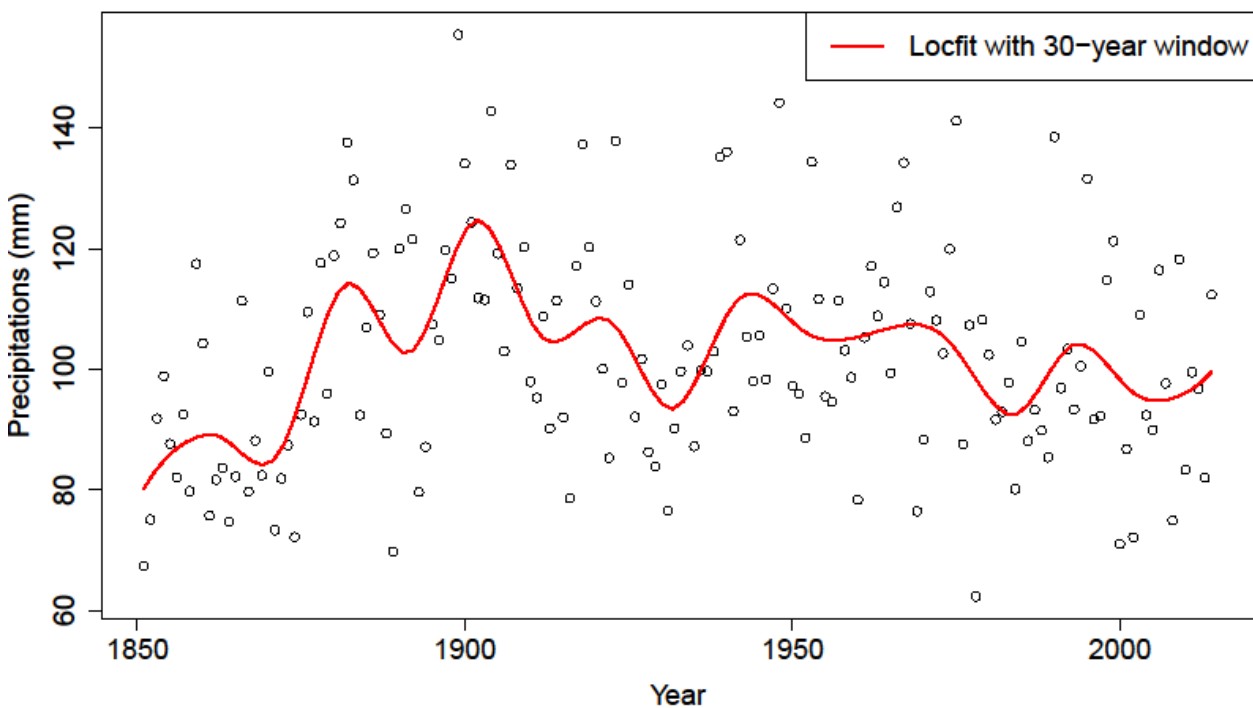

**Figure 1: Time series of annual maximum 30 day precipitation for the Highland Valley open pit copper mine in British Columbia, at latitude 50.49°N, and longitude 121.05°W, as estimated from the NOAA-CIRES 20th century climate re-analysis. This mine is one of the largest in the region, operating since the early 20th Century. The highest precipitation for any consecutive 30 day period in each year is shown, together with a 30 year moving window regression to illustrate the trend**

From a financial perspective, such regime like behaviour is of interest. For a given asset, if the pre-design period corresponded to a wet regime, infrastructure could well be over designed, and the associated capital cost may negatively impact the project's economics. Conversely, if this period was lacking rainfall extremes, and if the next 10 years are expected to correspond to the regime with a high frequency of extreme rainfall, then production losses and reconstruction costs, even using a modest discount rate, may have a much higher than anticipated impact on the valuation of the mining asset. For an insurance contract, this would correspond to a clustering of payouts over that period. This may also translate into higher insurance rates, which may not be reduced as a transition to the regime where the frequency of extreme events goes down occurs. Finally, in this scenario the investor may face a stranded asset, as the costs of reconstruction and liabilities caused by catastrophic failures may be prohibitive. Long climate records are needed to identify the temporal structure of the risk of extreme events, and to reflect it in subsequent risk analyses, so that appropriate estimates of the risk anticipated in the next decade or later can be made.



For the mining industry, depending on the duration and intensity of an extreme rainfall event, a cascade of direct and indirect financial impacts can result. These include:

- Production losses resulting from flooding of mine operations, loss of roadways, tailing dams, electricity services, equipment and/or housing
- Fines and clean up costs, due to release of pollutants from tailing dams and from the site into water bodies
- Increased costs related to dewatering procedures and new capital expenditure, pollution clean-up, or even impacts on ecology, human health, and casualties
- Increases in insurance premiums
- Increased regulation, design standards and associated costs
- Asset stranding if restarting operations may is financially, politically, or physically infeasible.

Some specific loss events may be insured, while others may not.

For example, in December 2010-January 2011, Queensland experienced heavy rainfall. This region has an active coal mining activity, with a complex system of mining assets and railways to bring coal to shipping ports. The event led to long dewatering processes, railway impairment, significant losses for mining companies and even spikes in coal prices (Chambers, 2011) (Regan, 2011). Overall, Queensland coal production missed its target by 40 million tonnes in 2011 (Heber, 2013), and Australian production decreased for the first time since 1981 (BP, 2016). This contributed to a record price of $330/tonne for hard coking coal (IBIS World, 2011) (Bloomberg, 2011) (Bloomberg, 2015).

In February 1994, a 31 m high tailings dam at the Harmony Gold mine in S. Africa failed due to overtopping following a heavy rainfall event (Van Niekerk & Viljoen, 2005). Nearly 300 hundred houses were swept away or damaged, and 17 people were killed. The subsequent investigation of the disaster led to a reformulation of policies, design standards, and monitoring requirements for tailing dams. These have had an impact on the subsequent cost structure for managing such risks, but to our knowledge have not triggered a significant evaluation of the methods used to manage the residual risk from extreme rainfall events in the industry, other than the purchase of limited liability property and casualty and business interruption insurance coverage.

The potential impact of extreme climate events is present even for mines not being actively operated, and may still exist for mines that stopped producing many years ago. Estimating potential damages associated with different levels of extreme hydrologic events at each mining site is difficult. Some companies are more financially exposed to these risks than others based on geography, exposure, and mitigation strategies.

Besides, for a portfolio that is composed of assets at many geospatial locations, one has to question whether the hydroclimatic risk factors are correlated across sites, such that many locations may experience floods or droughts in the same fiscal year, thus amplifying the impact of water and climate risks assessed for each site. A significant amount of research on the



geographical impacts of the quasi-periodic climate variations suggests that many regions in the world can experience persistent changes in risk, depending on the phase of a climate oscillation. These regions may or may not be spatially contiguous. For instance, during an El Niño or a La Niña phase of the El Nino Southern Oscillation, large parts of the world experience floods or droughts (Dai & Trenberth, 1998). A portfolio risk analysis of exposure to climate and water risks consequently needs to

integrate over both the space and time structures of climate to account for clustering in the exposure to these risks, rather than considering them to be independent in time and space.

Many companies have started commissioning consultant reports as to their carbon and water footprints, and more recently to their risk projected for various IPCC climate change scenarios to the year 2050 or 2100 (Rajczak, Kotlarski, & Schär, 2016). Unfortunately, the current generation of models of the coupled ocean-atmosphere circulation, i.e. global climate models, when

applied to the conditions of the 20th century, fails to reproduce the type of memory and oscillatory behaviour, as well as the spatial correlation structure that is noted in long observational records. Further, basic statistics (e.g., mean, standard deviation and skew) of hydroclimatic extremes tend to be strongly biased relative to the 20th century observations in most locations in the world (Woldemeskel, Sharma, Sivakumar, & Mehrotra, 2012). An industry focused on bias corrections of these statistics and the use of these corrections for future projections has evolved. A popular and potentially effective approach for

"correcting" such biases is quantile mapping (Rajczak, Kotlarski, & Schär, 2016), where the probability distribution of daily rainfall from an IPCC model for a historical period is scaled, quantile by quantile, to match the probability distribution of rainfall recorded as historical data at a particular location. This mapping is then extrapolated to the future period, a procedure whose reliability cannot be tested until the future occurs, since we don't know the source of the bias in the models used, and how that would propagate under extrapolation to a higher greenhouse forcing. Such point by point bias correction methods are

thus not able to address the biases in long term quasi-periodic evolution of climate, and do not constitute a reliable approach to future risk analysis since they represent a brute force attempt to correct and extrapolate selected output statistics, rather than addressing the deficiencies of the physics in the models. However, climate models are also applied to the 19th and 20th century conditions to build simulations called "re-analysis". The re-analysis models are very similar conceptually to the IPCC models used for future extrapolations, with one important difference. These models use data assimilation of observed surface

temperature and pressure records over the historical period. This means that the values of the climate variables computed by these models are updated to match as well as possible historical observations every single day. In effect, in this mode, the climate models are used to spatially interpolate the historical climate observations.

Observed data is sparser as one goes back in time, and during the world wars or other insurgencies, and hence the uncertainty and bias associated with the "re-analysis" reconstruction of the climate data fields varies as a function of time and space.

Nevertheless, the multiple sources of "re-analysis" data that are available for daily rainfall, temperature and other variables, can be very useful for portfolio risk analysis, since:



1) they provide a common period of global data coverage of 100 or more years (depending on which climate model is used);

2) as their temporal evolution is constrained by observations, they preserve the information on the phase of a climate oscillation across the world, thus providing information on the potential for spatial and temporal clustering of the frequency and intensity of hydroclimatic extremes;

3) they give the ability to assess how the hydroclimatic risk has evolved in the past in the best case scenario of an application of a climate model, thus providing a baseline against which future climate model based projections could be scored.

Consequently, while the procedures we develop here could readily use future climate projections, in this paper we choose to develop examples that use long historical data sets, so that we can reveal how potential changes in portfolio risk associated with rainfall extremes may have manifested over the past century or longer, thus providing a changing baseline for the risk that needs to be understood before undertaking an extrapolation to the future.

This paper represents the first attempt to develop an index for the exposure of a geographically diversified, global asset portfolio to the time-varying risk of climatic extremes using daily global rainfall data sets derived from climate re-analysis models. For the example presented here, we consider the mining sector, and extreme rainfall of specified duration as the risk factor. Once again, the analyses presented can be readily extended to consider the use of future climate projections based on the IPCC climate change scenarios. In this paper our emphasis is on exposing the potential for portfolio risk associated with climate risk, rather than the potential change of this risk as per these scenarios. The limited ability of these future scenarios to accurately inform extreme rainfall at this point of time, and the need to consider globally applicable uncertainty and bias correction methods to make these scenarios usable, leads us to consider that extension in a later paper.

The approach to the development of the index is described, followed by applications to selected sites and mining company portfolios. Extensions to other climate events, and to other applications, including simulation, value at risk analyses, and portfolio optimization are finally discussed. The functioning of a web based application has been developed to allow a user to conduct all the analyses described and illustrated in this paper.

## 2. Structuring a Risk Index for Climate Extremes

The risk associated with an extreme rainfall event depends both on its probability of occurrence, and on the potential financial impact. This latter includes direct operational loss to the company, as well as potential liabilities from harm caused to others. Yet, direct causality between climate events and issues at the mine site may be hard to quantify, as parameters such as infrastructure design, mining methods, acid consumption, or water management policies all play a role regarding the ways the impact of climate events is manifest. Such data is typically not available to people outside the mining company, and even for the company it may be difficult to estimate the extent of a projected loss from an extreme event.





Especially for rare or catastrophic events, it is difficult to develop a priori estimates of impacts, asset by asset, as they depend on the details of several site specific attributes, such as local demography and development level, or details of actual construction and monitoring of infrastructure, information on which may not be easy to develop. A well-run company may conduct a risk profiling exercise that identifies possible impacts contingent on certain types of events. An investor may indeed

ask for such disclosure, covering the events of concern and their estimated annual probabilities of occurrence. However, if such information is not available, one needs a consistent approach for scoring potential impacts, such that a fair index of exposure can be derived for a particular portfolio, whether it is composed of all mines in a particular geography, or a sector of mining, or belonging to a specific company. We develop such an approach here, and illustrate how the index derived can be used to:

1) understand the potential clustering of impacts in a sequence of years;

2) assess the impact of climate trends, production and price cycles on the exposure index;

3) compare the portfolios held by two or more companies;

In the examples considered in this paper, we define extreme events in terms of the T-year return level (level exceeded by the annual maximum with a probability of 1/T in any given year) based on available reanalysis datasets with at least 100 years of

data. Two candidate extreme events are considered:

- A 1-day annual maximum extreme rainfall event with a 100 year return period, i.e., an average 0.01 probability of yearly occurrence ($p = 0.01$), and
- A 30-day annual maximum rainfall event with a 10 year return period, i.e., an average 0.1 probability of yearly occurrence ($p = 0.1$).

The one-day extreme event is used as an example for rapid onset events that could induce spills and problems with tailings dams for a mine, while the 30-day event is used to consider events similar to ones that occurred in Queensland in 2010-2011, that are the consequence of persistent moderate to high intensity rainfall events over a long period. A specific 30 day extreme event may or may not include a 1 day extreme event, in a given year. A site can experience an event exceeding the target threshold several times during a given year as long as averaging windows (e.g., 30 days) do not overlap.

The motivation for the above choices is based in part on engineering design and regulatory practice, and in part on a desire to standardize exposure metrics. Depending on the type of mine or industrial installation, design guidelines for protection from flooding or extreme rainfall events typically refer to an event duration and an annual exceedance frequency corresponding to that duration. Thus, a holding pond for potentially contaminated runoff from rainfall on a site may be designed to hold the volume generated by a 30 day rainfall event with a 10 year, return period, while the main tailings dam may be designed to be

able to capture the volume of water generated by a nominal 100 year, 1 day rainfall event.





As was indicated in the introduction, given the short records typically used, there is considerable uncertainty as to the magnitude of the estimated 10 year or 100 year rainfall events at a site. Since climate statistics are not stationary, any given 30 year period of data used for such inferences may not be representative of the next 30 years when the business is operating. Since there is no easy way to know a priori which specific period of record (e.g., 1940-1960 or 1960 to 1975) was used for the

design of facilities at a particular mine or business site, it makes sense to refer the threshold to the longest period of record available to us, across all sites, and to then assess the space-time correlation and hence portfolio exposure to thresholds estimated across this entire period.

At each site, the nominal values corresponding to the extreme events of interest are computed from the NOAA-CIRES 20th Century Reanalysis V2c or the ECMWF ERA 20C reanalysis, which, to our knowledge, are the best precipitation datasets

according to our criteria (global coverage, relatively high resolution, and a long record) (Smith, Compo, & Hooper, 2014) (Dee, et al., 2014) (Irving, 2016). The NOAA-CIRES 20[th] Century Reanalysis V2c provides reanalysis rainfall data from January 1[st], 1851 to December 31[st], 2014 (NOAA ESRL, 2016) with a spatial resolution of 2° x 2°. The ECMWF-ERA 20C dataset provides daily precipitation data from January 1[st], 1900 to January 1[st], 2011, with a spatial resolution of approximately 125 km by 125km (NCAR UCAR, 2016). The NOAA-CIRES 20[th] Century Reanalysis V2c was downloaded from the NOAA-

ESRL website (NOAA ESRL, 2016), while the ECMWF-ERA 20C data was downloaded from the NCAR-UCAR climate data website (NCAR UCAR, 2016).

As previously mentioned, since the spatial density of observations varies over time, the precision or accuracy of the estimates by these models also changes. Further, precipitation is highly variable in space, and hence a model with even a 1.25° by 1.25° spatial resolution is too coarse to provide useful information as to extreme rainfall. This is definitely an issue, and motivates

our approach to look at the number of exceedances of a specified quantile at each location, rather than at the absolute magnitude of the rainfall generated by the model.

We expect that even the re-analysis models will be biased relative to at site observations. However, noting that quantile mapping for bias correction of the IPCC models is seen as an effective strategy, we expect that using the quantiles of the model precipitation at a given location to define the threshold of exceedance for extreme rainfall may provide a reasonable internal

self-consistency for the comparison of the relative exposure across different locations. Specifically, we assume that if the p[th] quantile of model based annual maximum precipitation is exceeded by $n$ days in a year at location $i$, and the p[th] quantile of model based annual maximum precipitation at location $j$, is exceeded by $m$ days at that location in a given year, then the relative magnitude of $n$ to $m$ exceedances at those two locations using the model based data is a good measure on average of the relative exceedances of the corresponding p[th] quantiles of observed annual maximum precipitation at the two sites. Recall

that we are using model based rainfall, since long records of observed rainfall at most of the sites (mines) of interest do not exist. While these long model based records may not get the rainfall statistics at a given site exactly right, the persistence of



extreme wet or extreme dry conditions across a region, or across a historical period is likely to be connected to features of the large scale circulation of the atmosphere, which the models are expected to resolve quite well. Thus, for our purpose of exploring the spatial and temporal correlation of the risk of extreme rainfall event exceedance across many sites in a portfolio, and the relative risk of exposure of portfolio A to portfolio B, the approach chosen may be satisfactory. Uncertainty due to the

model structure and to the data assimilation strategies can be explored by using multiple re-analysis models, and the ones used in this paper are the ones that as of the date of publication, provide the longest re-analysis climate records.

## 3. Approach

Given the discussion in the previous section, we consider an event that triggers possible financial exposure of concern at a given site to be indexed to the exceedance of the p$^{\text{th}}$ quantile of annual maximum rainfall of a duration of $d$ days. Depending

on the investigator's interest, one can consider exposure relative to specified values of $p$ and $d$ at each site, the direct and indirect financial exposure to each such event at each site, and aggregate the exposure across sites, for each year of the historical record to provide a time series of the index of exposure for the portfolio of interest. Time series of the index can then be analysed for cyclical or secular trends, evidence of spatial clustering, and the relative value at risk for portfolio A vs portfolio B. The entire exercise could be repeated for different combinations of $p$ and $d$ to assess the kinds of events that may lead to

the most differences in relative exposure. These ideas are developed formally below, and a web app that implements the analyses is available from the authors.

### 3.1 Climate risk exposure

We first choose a fixed level of exposure for a given climate event (extreme rainfall or drought) expressed in terms of its nominal annual probability of occurrence p over a year. To explore the space-time structure of exceedances of this threshold,

we first identify the exposure frequency of the event at a given asset, for each year of the historical record.
A first step is to identify the annual maximum of rainfall for duration $d$ at location $i$ from the climate data set, for each year of the record. This time series of annual maxima is denoted as $Precip_{it}^{max}$. The p$^{\text{th}}$ quantile $Precip_{i}^{max,p}$ of $Precip_{it}^{max}$ is then estimated as the empirical quantile or after fitting a Generalized Extreme Value distribution (GEV) (Katza, Parlangeb, & Naveauc, 2002) to $Precip_{it}^{max}$.

Let's call $X_{j}$ the cumulative rainfall over $d$-days for a window starting on day $j \in [\![1, J]\!]$ of the year (in practice, $J = 365 - d + 1$ for a regular year, $J = 366 - d + 1$ for a leap year). Then, under the assumption that the $\{X_j\}$ random variables are independent and identically distributed (i.i.d.), the Extreme Value Theorem (Coles, 2001) tells us that calling $M_n = \max_n(X_j)$, if there exists sequences $(a_n) > 0$ and $(b_n)$ such that $M_n^* = \frac{M_n - b_n}{a_n}$ converges to a non-degenerate cumulative distribution function (cdf) $G$, $G$ is of the GEV family i.e. $G$ can be written as:





$$G(z; \mu, \sigma, \xi) = \exp\{-\left[1 + \frac{\xi(z-\mu)}{\sigma}\right]^{-\frac{1}{\xi}}\} \quad (1)$$

where:

μ is the location parameter,

$\sigma > 0$ is the scale parameter,

$\xi$ is the scale parameter.

This distribution family can be divided into three sub-families:

- for $\xi > 0$, it is of the Fréchet type,

- for $\xi = 0$, (limit of (2) when $\xi \to 0$) it is  of the Gumbel type,

- for $\xi < 0$, it is of the Weibull type.

This theorem is the counterpart of the extreme limit theorem for block maxima, and is in practice used in a similar way: to find an approximate distribution of $M_n$ for $n$ large enough (in practice, $n$ is fixed). The existence of the  $(a_n)$ and $(b_n)$ sequences is assumed, and while these numbers are unknown, as, for $n$ large enough, $\Pr(M_n < z) \approx \mathrm{G}\left(\frac{z-b_n}{a_n}\right) = G_0(z)$, where $G_0$ is a distribution of the same family, then the cdf of $M_n$ can be approximated (Coles, 2001). Obviously, critical assessment of the model fit needs to be performed, as with any statistical inference.

In practice, the cdf of $M_n$ is thus estimated by fitting a GEV to the $Precip_{it}^{max}$ time series obtained from data. While the i.i.d. assumption does not hold in our case (moving window precipitation totals are obviously not independent), adjustments to the location and scale parameters can account for the time clustering (Katz, 2013); the process is still valid if the $\{X_j\}$ are of the same family (which takes care of the fact that the distribution parameters might vary depending, for instance, on seasonality). The GEV model can account for non-stationarity by making μ, σ and/or ξ functions of time $t$, although allowing ξ to vary

generally makes convergence difficult. Then, $\mu(t)$ will describe trends and cycles of the center of the distribution, while σ(t) will describe evolutions of the "size" of the deviations about μ (Katz, 2013). At a mine, study of the time series $Precip_{it}^{max}$ through GEV can enable one to understand if and how exposure at different return periods has changed over time and what consequences this can have relative to the infrastructure design. Confidence intervals of such return levels can also be estimated. The parameters , $\mu(t)$ and σ(t) can be estimated using maximum likelihood, and different forms of time variation

(including constant for stationarity) of these parameters can be explored, and the best model selected using the BIC criterion (Katz, 2013).

Next we can develop a time series of physical exposure at each mine $i$, by estimating $n_{i,t}^{p}$ as the number of events of duration $d$, that exceed the stationary threshold $Precip_i^{max,p}$ at mine $i$ in year $t$, based on the long run data (or for a mine operator who has information on the original data used, the data period used for design). Then the statistic

$$N_t(p) = \sum_i n_{i,t}(p) \quad (1)$$

can be used to get insight into risk exposure at the portfolio level.



The degree of clustering of exposure across mines in the portfolio, and whether there are temporal trends or cycles in such exposure can then be investigated using $N_t(p)$. Clustering can be assessed by comparing the probability distribution of $N_t(p)$ against what would be expected under independence of occurrence of extremes at each mine, and trends can be assessed via standard methods of trend and cyclical analysis.

**3.2 From climate exposure to financial risk**

In the financial industry, a common measure of risk is the Value-at-risk or VaRq. It is defined as the potential loss (incurred by a given risk factor) over a certain time period that won't be exceeded with a given confidence level q (Webby, et al., 2007) (Yamout, Hatfield, & Romeijn, 2006) (Adriaens, Sun, & Gao, 2014). A Conditional Value at Risk (CVaRq) is defined as the expected value of the loss in case the VaRq is exceeded. We take an approach consistent with these ideas, while recognizing

that it may not be easy to estimate the direct loss associated with different levels of events at each mining site.

Our strategy is to use a weighting of the $n_{i,t}(p)$ time series to be able to compare portfolios to each other, rather than estimating actual VaRs, based on mine by mine potential loss associated with the threshold event. For each mine site $i$, a potential loss $L_i(p)$ is associated with the event with return period p. We assume that $L_i(p)$ can be decomposed as:

$$L_i(p) = C(p)V_i + D(p)F_i \qquad (3)$$

where:

- $C(p)$ and $D(p)$ are constants associated with the rarity of the event, that apply to direct loss and external loss respectively

- $V_i$ is a measure of the financial value of the mine

- $F_i$ is a measure of the potential value of impacts on society external to the mine, that the mine owner is liable for.

$V_i$ can for instance be the production rate of the mine, some multiple $C(p)$ of which may be lost due to disruption, for an event

with a probability of occurrence p in a given year. Alternately, $V_i$ could be the estimated NAV of the mine, which may be relevant as a measure of the scale of the asset at risk. Production loss could be used in the context of an event expected to incur mine flooding, difficulty of access, or cut in production due to drought. In such a case, $L_i(p)$ would represent the value of the potential loss of production due to disruption, and one would expect that as p decreases (the event is more extreme), $C(p)$ increases. Similarly, NAV could be used to reflect potential closure of the mine, or a long term suspension of operations

due to a catastrophic event. The probability p considered for an index that uses this measure for defining $V_i$ would logically be lower than the ones used for a production based index.



Correspondingly, one can develop arguments for $D(p)$ and $F_i$ considering the population or ecosystems that are likely to be affected as a consequence of the failure of systems at a mine in the event of extreme rainfall. This could include environmental impacts, health impacts, loss of water services to a community, and/or the financial impact from mine closure. Available satellite remote sensing and geospatial data bases provide information on hydrologic networks, population density, GDP and

ecological attributes that could be identified downstream of each mine, and used to parameterize $F_i$. In reality, it is difficult to develop estimates for $D(p)$, $C(p)$ and $F_i$ without insider information. Therefore, in the examples developed in this paper, we take $C(p)$ to be 1, and $D(p)$ to be 0.

Effectively, we assume that there is a valuation associated with the mine as well as with the potential area of external impact, and that for an extreme event of a specified rarity (probability of occurrence), across sites, the loss is proportional to that

valuation. As an example, if a 100 year event (p =0.01) were to occur at two mines, with the market apportioning $10 million to one asset and $100 million to another asset, in the absence of other information, we are assuming that the financial impact is directly proportional to the relative valuations attributed by the market to each asset. This implies that for a 100 year event that results in permanent mine closure (for example), the resulting impact on the company's valuation as a result of the event at the second mine would be 10 times greater than the same event at the first. While this is unlikely to be an exact measure of

financial impact, it represents a relative measure of exposure, and hence provides a basis for developing a comparable index across a portfolio. Where detailed information on the potential asset level loss probability distribution and the direct resulting financial impact is available, it would obviously be better to use it directly. Varying $p$ allows for the development of a probabilistic risk profile across a portfolio. One can see that for a given $p$, the contribution to the expected value at risk can be readily evaluated, under assumptions of a stationary climate, as $pL_i(p)$.

We can then define portfolio level financial exposure as,

$$S_t(p) = \sum_i L_i(p)\, n_{i,t}(p) \qquad\qquad (4)$$

which can be reframed as

$$S_i'(p) = \sum_i V_i\, n_{i,t}(p) \qquad\qquad (5)$$

since $C(p)$ is assumed to be a constant across all assets, and for now we assume that we are only considering direct impacts to

the mine. Normalizing by the portfolio valuation, we define:

$$R_t(p) = \frac{S_t'(p)}{\sum_i V_i} \qquad\qquad (6)$$





which provides a metric for the relative volatility or risk exposure of different portfolios (companies or economic sectors), normalized by their valuation. For instance, two different companies can be compared in terms of the quantiles and trends of their respective $R_t(p)$. Varying $p$ also enables the exploration of the variations in tail risks of a given portfolio.

For a specified annual probability of exceedance p, considered to be the design level for infrastructure at the mine, the q[th] quantile, $S'_q(p)$ of $S'_t(p)$ can be considered to be a measure of the VaRq for a mining company, and the corresponding q[th] quantile $R_q(p)$ of $R_t(p)$ provides a scaled measure that allows a comparison of the VaRq exposure of each company as a fraction of their total production or total portfolio value.

Finally, we can define a measure similar to CVaRq for the potential expected loss in case an event with a probability lower than p occurs with a probability (1-q) of the time as:

$$CVR_q(p) = \frac{1}{1-q} \int_{R_q(p)}^{R_m(p)} R_t(p) f(R_t(p)) dR_t(p) \tag{7}$$

This is numerically evaluated as:

$$CVR_q(p) = \frac{1}{(1-q)(m+1)} \left\{ \frac{R_q(p) + R_m(p)}{2} + \sum_{k=q+1}^{m-1} R_k(p) \right\} \tag{8}$$

Where $m$ is the number of years in the record, and $R_k(p)$ represents the k[th] ranked value of the series $R_t(p)$, such that $R_q(p)$ corresponds to the q[th] quantile of $R_t(p)$.

Similarly, one can define

$$CVS'_q(p) = \frac{1}{1-q} \int_{S'_q(p)}^{S'_m(p)} S'_t(p) f(S'_t(p)) dS'_t(p) \tag{9}$$

Further, such a procedure can be used to generate inputs for real option analysis models to inform the Value-At-Risk (Blanchet & Dolan, 2016). For different values of  p, distributions of $n_{i,t}^p$ , $N_t(p)$, $S'_t(p)$ or $R_t(p)$, can be used to simulate extreme event impacts.

## 4. Example applications – frequency of events at the mine level

We start with an example for the analysis of the $Precip_{it}^{max}$ time series for a given site using GEV distribution for threshold selection. Consider again the Highland Valley open pit copper mine in British Columbia. Let's consider a 30-day event with a return period of 10 years. Using the NOAA-CIRES 20th Century Reanalysis V2c dataset, we develop the yearly maximum 30-day precipitation time series $Precip_{it}^{max}$. The eXtremes package in R (Gilleland, 2015) was used to estimate an appropriate



parametric model in the GEV framework associated with this annual maximum time series. For the stationary assumption, the 10 year event is estimated as 126.2 mm consistent with the empirical quantile of 126.2 m, with a 95% confidence interval of [122.21 mm; 131.19 mm].

We considered polynomial models in time of order 0 to 4 for both the location and scale parameters of the GEV distribution,

leading to 24 models to be tested, including the stationary model. The best model based on the BIC criterion is the model using a quadratic model for the location parameter and a constant for the scale:

$$\mu(t) = 74 + 0.47\ t - 0.0039\ t^2 ; \qquad \sigma = 13.4 ; \qquad \xi = -0.26 \qquad\qquad (9)$$

A likelihood-ratio test between the stationary model and this model leads to a p-value of 4.85e-05, thus enabling us reject the null-hypothesis of no trend. Standard diagnostic tests support the applicability of the non-stationary GEV model. The return-

10 level plot on Fig. 2 shows the effective return-level plot of a 10-year, 30 day rainfall event for the non-stationary model in blue under stationarity assumption, and for the model selected thanks to BIC in red. A nonparametric trend function, as illustrated in Fig. 1 could potentially reveal additional structure. If a more detailed characterization of the decadal variations in the return level were of interest at this site, the use of a spline basis function for the trend in the location parameter may be appropriate, as shown by (Bocci, Caporali, & Alessandra, 2012) (Padoan & Wand, 2008) (Nasri, El Adlouni, & Ouarda, 2013) (Yousfi &

El Adlouni, 2016).

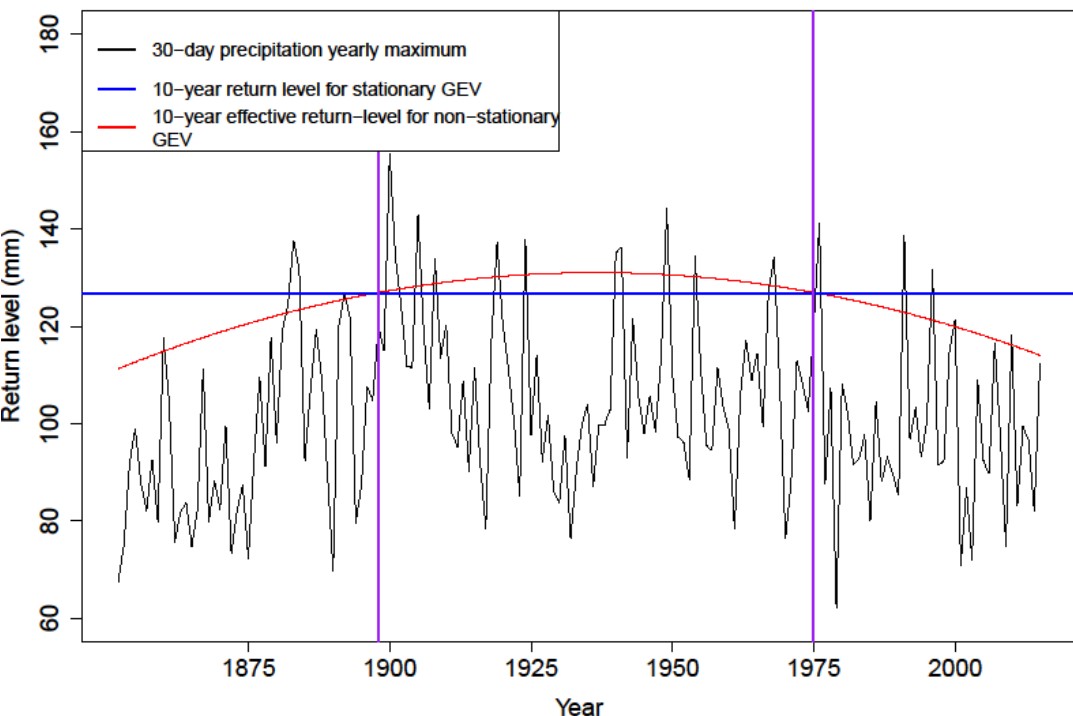

**Figure 2: Effective 10-year return level according to the non-stationary GEV distribution for the Highland Valley Mine**



## 5. Example applications – portfolio level

In this paragraph, we provide three examples at the portfolio level across a set of mining companies for which we have information on asset locations and $V_i$.

1) The purpose of the first example is to explore whether spatial and temporal correlation in the frequency of climate extremes leads to portfolio tail risk that may a) be substantially greater than expected from treating each asset as an independent exposure, b) have systematic increasing or decreasing trends or persistence. For this case, we study the number of events affecting a given portfolios of mines via the corresponding $N_t(p)$.

2) The second application highlights how different choices for $V_i$ can be used to provide insight into financial exposure. Two weighting procedures are considered. One uses 2015 production and 2015 average commodity prices to attribute a value to each mine, and the other uses Net Asset Value from broker reports. In the first case, for mine i,

$$V_i = \sum_c P_c Q_{i,c} \tag{7}$$

where,

$P_c$ is the average 2015 unit price of commodity c obtained from (Word Bank, 2016), and

$Q_{i,c}$ is the quantity of this commodity produced by mine i in 2015.

In the second case, for mine i,

$$V_i = NAV_i \tag{8}$$

where $NAV_i$ is the Net Asset Value attributed to site i in the broker report chosen.

3) The third example highlights how the $N_t(p)$, $S'_t(p)$ and $R_t(p)$ time series can be used to compare the tail risk that results from clustering for two portfolio of mines, depending on how the assets are valued or grouped.

Data on different mining companies was gathered for these applications. The details of this data are provided in Appendix A. For the first application, we build portfolios of producing mining assets for four companies (BHP Billiton, 2016) (Rio Tinto, 2016) (Barrick Gold Corporation, 2016) (Newmont Mining Corporation, 2016) using their annual reports. Using BHP and Rio Tinto enables us to test our method on two large portfolios, to measure whether or not their portfolios are diversified with respect to the risk of rainfall extremes. Barrick Gold and Newmont Mining, two of the main gold miners, are chosen because they are similar in terms of size and business, and can be also used to illustrate applications 2) and 3). In this case, a mining asset refers to a unique physical site. The data set includes mines that may be listed as "on care and maintenance" but excludes undeveloped projects. Generally speaking, mining portfolios were disaggregated based on public disclosure using our best judgement on what constitutes an individual asset, for each mining company, and the share of ownership in jointly owned assets by multiple companies.

For the second application, the portfolios of mines defined for Barrick Gold and Newmont Mining for the first application are used first. These two companies produce mainly gold and copper, and the reported production of these two commodities is used to value each mine as indicated in equation (5). Then, we consider NAV weighting for Barrick Gold and Newmont,





obtained from broker reports (TD Securities, 2016 a.) (TD Securities, 2016 b.). These portfolios of mining sites are different from the ones defined for the first application: they are composed of the assets valued in the broker reports used to obtain NAV valuations (whether they are undeveloped projects, producing assets, or even closed mines).

In total, for the first two examples, 6 different portfolios of mines are considered (one for BHP and Rio Tinto, and two for
Barrick and Newmont). For the second application, Barrick Gold and Newmont Corporation are compared using the different weighting methods proposed ($V_i = \sum_c P_c Q_{i,c}$ , $V_i = NAV_i$).

For the last application, portfolios of mine sites of 15 companies for which we have asset-level NAV valuations from TD Securities broker reports are used (including the ones for Barrick Gold and Newmont Corporation already introduced for the NAV weighing example in the second application) (TD Securities, 2016 a.) (TD Securities, 2016 b.) (TD Securities, 2016 c.)
(TD Securities, 2016 d.) (TD Securities, 2016 e.) (TD Securities, 2016 f.) (TD Securities, 2016 g.) (TD Securities, 2016 h.) (TD Securities, 2016 i.) (TD Securities, 2016 j.) (TD Securities, 2016 k.) (TD Securities, 2016 l.) (TD Securities, 2016 m.) (TD Securities, 2016 n.) (TD Securities, 2016 o.).

### 5.1 Frequency of events across a portfolio, $N_t(p)$

We consider a 1-day rainfall event with a 100-year return level and a 30-day rainfall event with a 10-year return level. For
each of the aforementioned companies, we compute the $N_t(p)$ corresponding to the portfolios of producing mining assets, using both the NOAA-CIRES V2c and the ECMWF ERA-20C climate datasets.

### 5.1.1 Trends and clustering in time

From studying the $N_t(p)$ for the four mining companies we find that:

- Statistically significant trends for increasing frequency are observed in most of the cases, in particular when using
the longer NOAA-CIRES V2c climate dataset.
- there is a cyclical behaviour regarding the number of exceedances of the thresholds defined.

In nearly all cases analysed (independent of the climate dataset used), we observe a cyclical behavior in the number of exceedances.  Figure 3 show the location of the mining assets for Rio Tinto. Figure 4 provides the time series of the yearly number of 30-day extreme events across this portfolio that exceed the 10 year return level at each site computed using the
NOAA climate dataset. For this case, we observe a high number of events for the periods 1940-1950, 1980-1990 and 1995-2005, while the 1950-1980 period is relatively quiet. Thus, infrastructure designed and constructed using the 1940-1950 record as a basis might have given this company's executives a sense of security during the 1950-1980 period, while the following years might have appeared as a period of high exposure. People tend to weight recent history more than the past, which would yield cyclical investment and attention to risk management for such a company. Similar results for other companies are shown
in Appendix B.





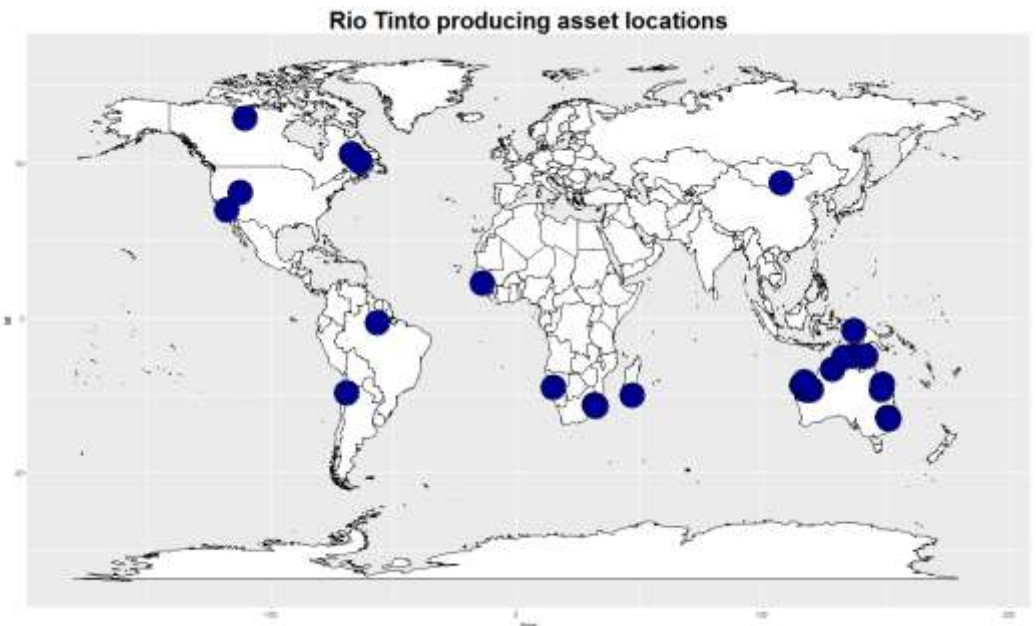

**Figure 3: Locations of productive Rio Tinto mining assets at the end of 2015 (some assets overlap on the map)**

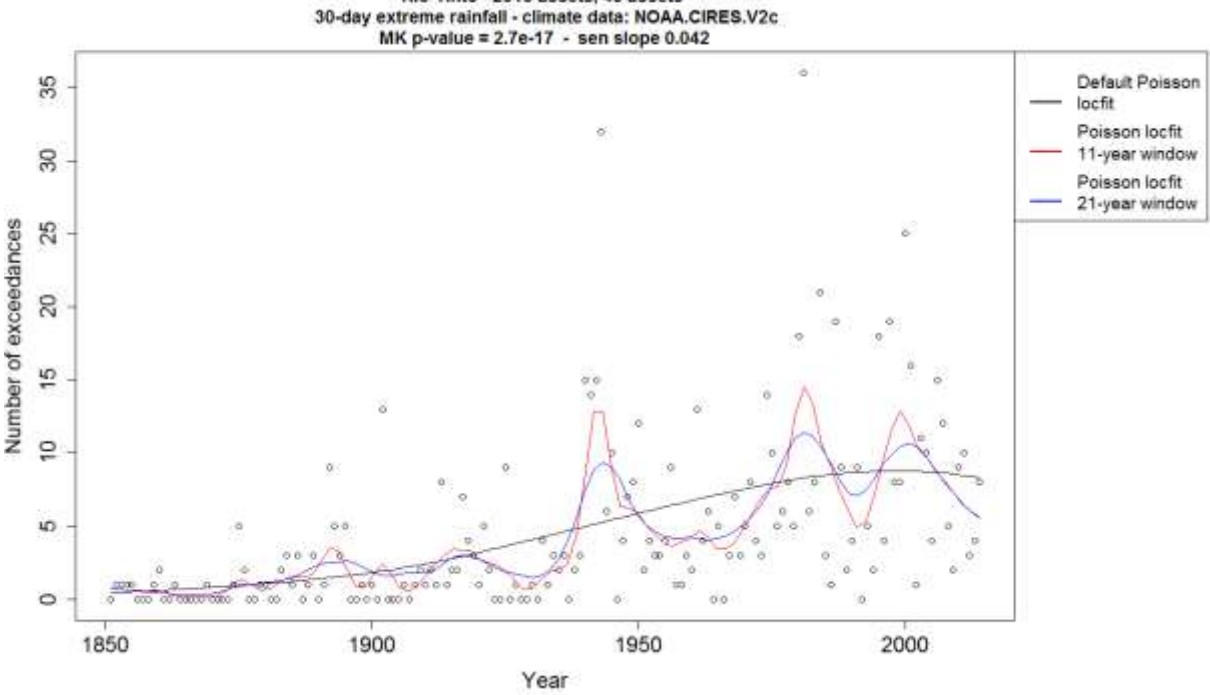

**Figure 4: Time series $N_t(p)$ of the yearly number of 30-day extreme rainfall events exceeding the 10 year return level for the Rio**

5    **Tinto portfolio computed using the NOAA-CIRES 20th Century V2c dataset, using 3 windows (11, 21, and 114 years) to illustrate different aspects of long term variability**



Significance of the trends for each of the $N_t(p)$ time series estimated was assessed using the Mann-Kendall test (Helsel & Hirsch, 2002) for monotonic trend. Results are provided in Appendix B. For the 100 year 1 day rainfall event, the number of events exceeding the design level at the portfolio level demonstrates statistically significant (at the 5% level) trends for all companies using the NOAA-CIRES data, and for Barrick Gold and Newmont using the ECMWF data. However, the trends

computed are null, which is due to the fact that a Sen Slope is computed as the median of the slopes between all the points in a dataset, and most of the years then correspond to zero values. For the 10 year 30 day event, the portfolio counts exhibit statistically significant upward trends for all mining companies when using the 164 year long NOAA-CIRES data, and only for Rio-Tinto when using the 111 year long ECMWF data. Thus, there is evidence for an increasing frequency of portfolio level exposure for both the more catastrophic short duration event and the long duration, more moderate event that we

hypothesize is related to persistent production disruption.

### 5.1.2 Clustering in space and time

Our key finding is that for all cases, the number of exceedances for each mining portfolio in many years is substantially higher than what would be expected by chance. There is evidence of very fat tails for the portfolio risk.

Representative results are discussed here, with all results presented in Appendix C. From Fig. 4, it is interesting to note that

there were 36 exceedances of a 10 year 30 day rainfall event were experienced in a portfolio of 40 Rio Tinto assets in 1981. We emphasize that as a single asset can potentially have multiple, distinct 30 day periods that can experience an exceedance of the 10 year event at the site in a given year. Thus, in the worst years, the number of exceedances may exceed the number of sites in the portfolio. For example, in 1981, several Rio Tinto assets were hit twice. These include the iron ore mines of the portfolio located close to each other in the Pilbara region of Australia. However, hits happened in various parts of the world,

and 22 different sites were hit (their geographic distribution is showed on Fig. 5).



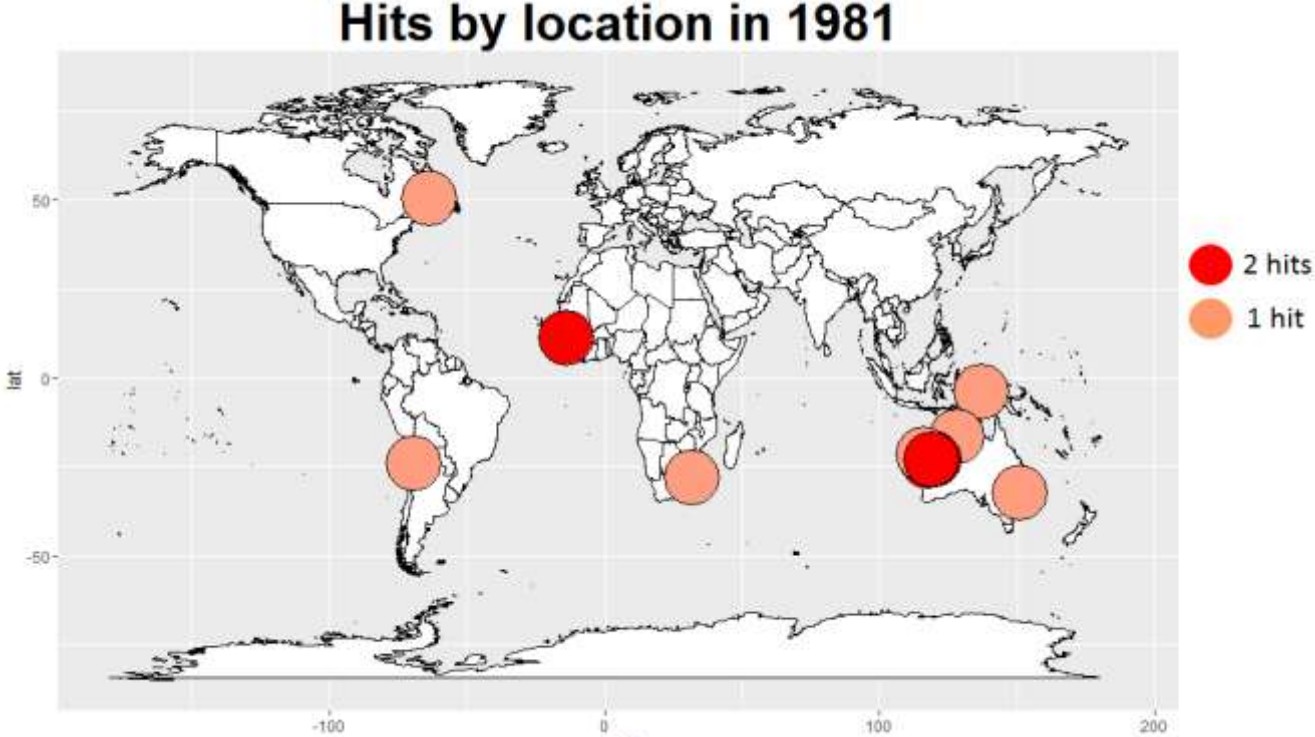

**Figure 5: Hits by location for the Rio Tinto portfolio in 1981 for a 30-day, 10-year rainfall event according to the NOAA-CIRES 20th Century V2C dataset (dark red corresponds to two exceedances, light red to one)**

Since on average one would expect 4 such exceedances (p=0.1 × 40) in a year, 36 hits is truly a remarkable number, suggesting
a very fat tail exposure for the portfolio compared to what could be expected by chance. Practically, if each such event were
to lead to even a 12.5% production loss (based on the 1 month duration of the event and 0.5 months to restore full production)
at a mine on average, then the portfolio would suffer a production loss of 35/40×12.5 =11% for the year, compared to ~1.2%
if there was no clustering across mines and in time. The financial impact for the mining company would depend on the fixed
costs that would need to be incurred irrespective of production (in the event of a production stoppage) as well as the foregone
revenue during the production stoppage. We note that there are other years in which very high counts are also recorded.
Consequently, it is useful to formally test whether or not, the number of exceedances across sites could occur if the climate
risk exposure across sites were random and independent.

For this check, we compare the empirical cumulative distribution function (cdf) of the data, $F(N_t(p))$ with the cdf that one
may expect if the underlying process that generated $N_t(p)$ were an independent and identically distributed (i.i.d.) process,
across the sites in the portfolio. As we have defined extreme events in terms of an yearly probability of occurrence $p$, at each
site, the theoretical process can be assumed to be a Poisson process with $\lambda = p$. Therefore, for m sites, under the assumption
of an i.i.d process, the theoretical distribution would be one of a Poisson process with $\lambda = mp$. For the Rio Tinto portfolio, the



number of exceedances with $\lambda = 4$ would be (9,11,13) for probabilities of (0.01,0.001,0.0001) respectively. Thus under the independence hypothesis there is a near 0 probability of 36 exceedances in a year, and in 15 out of 164 years, the number of exceedances is greater than 13, suggesting a very high incidence of clustering indeed. From Fig. 6, we note that for the 100 year, 1 day rainfall event, depending on the climate data set used, the number of portfolio events of concern at the 99th quantile

5    is 5 to 6 times what may be expected by chance for BHP Billiton, and 2 to 3 times what is expected by chance for other companies and quantiles. Similar results for the 10 year 30 day event are presented in Appendix C.

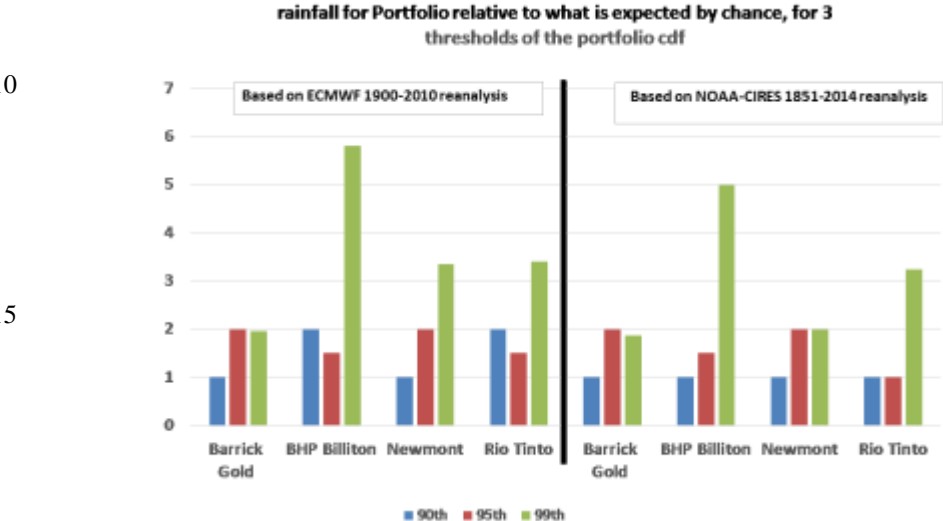

**Figure 6: Ratio of actual number of events in excess of at site 100 year 1 day extreme rainfall for a portfolio relative to what is expected by chance based on the Poisson distribution, for 3 thresholds of the portfolio cdf**

To the extent the market tends to look at each mine as an independent profit centre, generating value as a standalone entity and with a subset of risks that are independent to the other mines that it owns (and others such as commodity risk which are present across the entire portfolio), our analysis demonstrates that the exposure to extreme rainfall events needs to be looked at across assets; clustering is a significant issue and high impact events at the portfolio level may have a much higher probability of occurrence than anticipated under the usual analytical mindset.

**5.2 Extreme events and indexing of potential financial loss at the portfolio level**

In this section, we explore how the financial exposure of mining companies may be manifest for the two types of rainfall

30    events considered at the portfolio scale, by weighting the event occurrence with an appropriate financial variable. We focus on Barrick Gold and Newmont Corporations, two major gold miners, because these two companies are similar in terms of their core business, their diversity in geographic distribution of asset locations, and their revenues. We consider both a 1-day extreme





rainfall event with a 100-year return level, and a 30-day extreme rainfall event with a 10-year return level. All the computations in this section were performed using time series built with the NOAA-CIRES 20th Century V2c dataset, as it has a longer record, with 164 years of data.

We use two weighting methods:

- one that values each mine using an estimate of its recent annual production value,

- one that values each mine according to a recent Net Asset Value indicated in a broker report.

The method using production value is a measure of shorter-term impact of the events, while the NAV method may be used to measure more catastrophic losses.

### 5.2.1 Weighting with production

First, we develop an index using annual production data at the mine level reported in (Barrick Gold Corporation, 2016) (Newmont Mining Corporation, 2016). We focus on the two main commodities reported by these two companies: copper and gold. We associate to each mine its production multiplied by the average 2015 price of the corresponding commodity, therefore obtaining an estimate of the mine's 2015 revenue. Commodity prices (in nominal dollars) were taken from (Word Bank, 2016). While the total sales revenue mentioned in the 2015 annual reports of Barrick Gold and Newmont amount to USDm 9,029 and

USDm 7,729 respectively, the estimated values based on this indexing procedure are USDm 7,738 and USDm 6,240, which have approximately the same ratio.

Using this weighting method, we then analyze the tail exposure through the weighted time series $R_t(p)$. Figure 6 below shows the annual exceedance probability of exposure given by the $R_t(p)$ obtained for the Barrick Gold and Newmont Corporation portfolios both the 1-year, 1-day event and the 10 year, 30-day extreme event. Note for instance that for the 30-day event,

for Barrick Gold, over 99% (46%) of the company's total production value is exposed with a probability of 1% (5%) per year, while for Newmont the corresponding numbers are 90% and 58% (recall that the total can theoretically go beyond 100% has multiple hits can concern one asset in a given year). If we consider a 12.5% disruption in production due to each such event, then for Barrick and Newmont, the annual financial impact could be as much as $9×0.125×0.99 = $1.1 Billion, and $7.7×0.125×0.9=$0.87 B with a 1% chance in a given year.





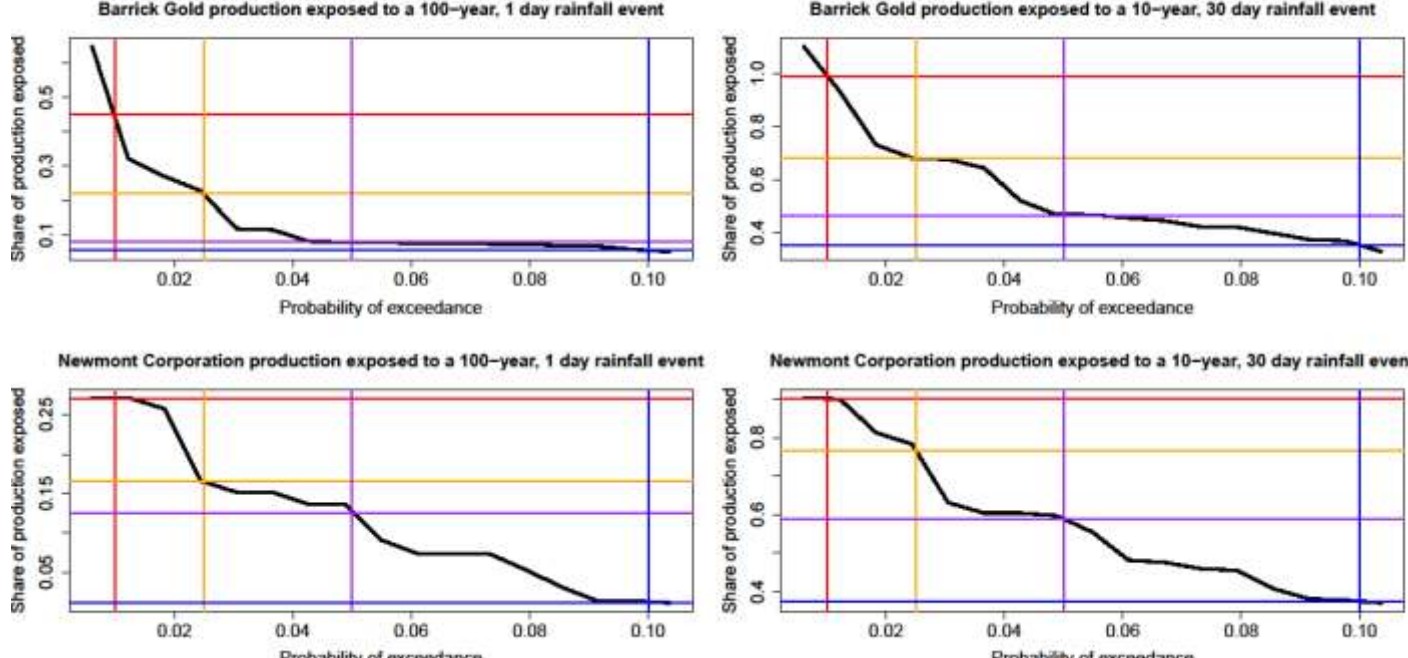

**Figure 7: Fraction of company production value exposed, as a function of annual exceedance probability for 100 year, 1 day and 10 year, 30 day events for Barrick Gold and Newmont based on the NOAA-CIRES data**

### 5.2.2 Weighting with Net Asset Values (NAVs)

5 For this example, we chose two reports from TD Securities written a few days apart: (TD Securities, 2016 a.) (TD Securities, 2016 b.). In the following, a mine or a project is included as long as it appears in the report. This leads to 19 sites mentioned for Barrick Gold, and 12 for Newmont Corporation.

In a similar analysis than in 4.2.1., note for instance that for a 1-day event, for Barrick Gold, over 29% (7%) of the company's total production value is exposed with a probability of 1% (5%) per year, for the while for Newmont the corresponding numbers

10 are 33% and 9%. If we consider a 10% destruction of value (which is likely a low number) due to each such event, then for Barrick and Newmont, the annual financial impact could be as much as $9×0.10×0.29 = $0.26 Billion, and $7.7×0.10×0.33=$0.25B with a 1% chance in a given year.





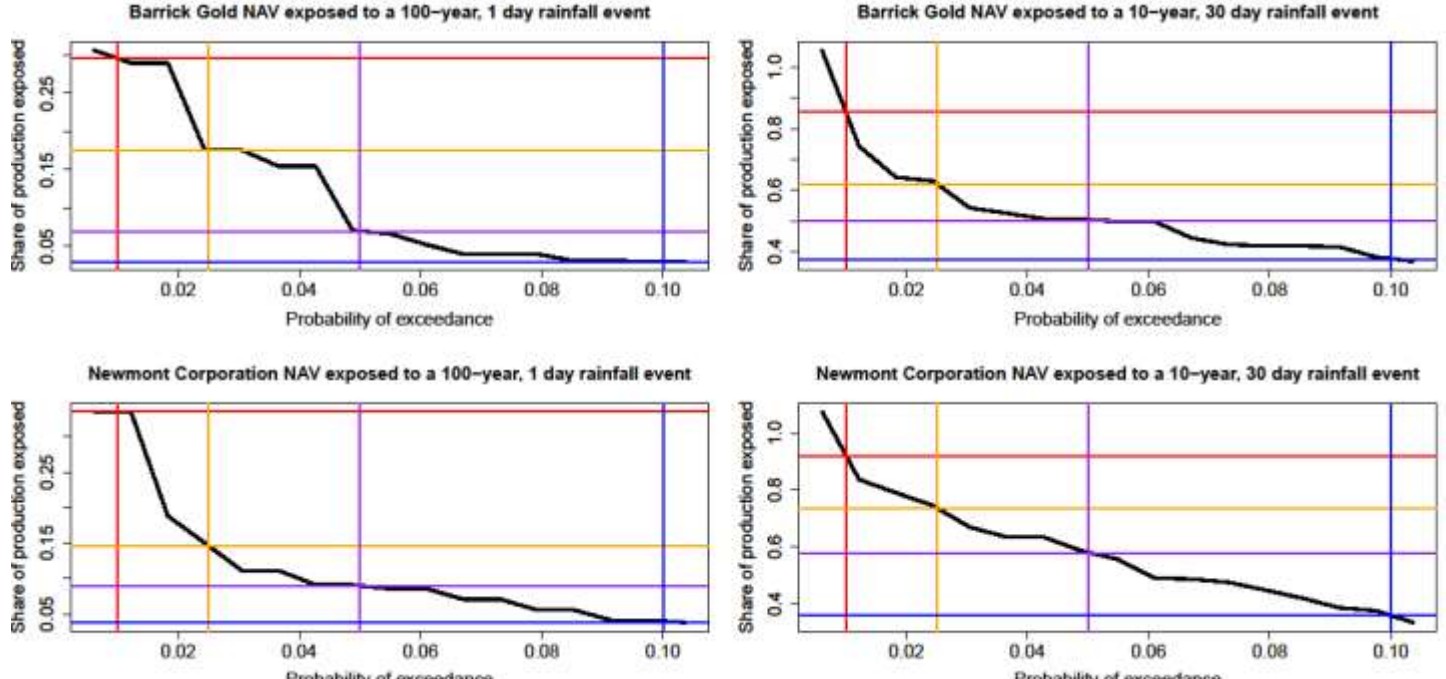

**Figure 8: Fraction of company value exposed, as a function of annual exceedance probability for 100 year, 1 day and 10 year, 30 day events for Barrick Gold and Newmont based on the NOAA-CIRES data**

### 5.3 Company comparison

5    We obtained asset-level valuation for 15 companies from TD securities, performed around the same time (Winter-Spring 2016) and present here the rankings ensuing from applying the method described using time series built using the NOAA-CIRES V2c climate data. We consider the 100 year, 1-day extreme rainfall with a 100-year return level, with NAV weighting and choose a portfolio tail-exposure level of q=0.95 to exemplify how the $S'_q$, $R_q$ and corresponding $CV_q$ measures that are similar to $VAR_q$. or $CVAR_q$ can be used to compare these companies exposure. A user may want to vary q, using our web app to

10    develop customized results (in the following, a higher rank means a higher potential exposition).

Table 1: Ranking of 15 companies based on $S'_{0.95}(0.01)$ and $CVS'_{0.95}(0.01)$ measures for a 1-day rainfall event, obtained using the NOAA-CIRES 20th Century reanalysis dataset and mine valuation obtained from broker reports from TD Securities

| Company | $S'_{0.95}(0.01)$ (USDm) | Rank $S'_{0.95}(0.01)$ | $CVS'_{0.95}(0.01)$ (USDm) | Rank $CVS'_{0.95}(0.01)$ |
|---|---|---|---|---|
| Agnico Eagle | 272 | 8 | 8.05E+02 | 8 |
| B2Gold | 204.3 | 10 | 4.14E+02 | 13 |
| Barrick Gold | 974.15 | 2 | 2.63E+03 | 2 |
| Capstone Mining | 0 | 14.5 | 1.10E+02 | 15 |




| | | | |
|---|---|---|---|
| Eldorado | 374.66 | 7 | 7.25E+02 | 10 |
| First Quantum Mineral | 136.595 | 12 | 3.65E+03 | 1 |
| Franco Nevada | 452.015 | 6 | 7.66E+02 | 9 |
| Goldcorp | 669.6 | 4 | 1.42E+03 | 4 |
| Hudbay | 0 | 14.5 | 5.58E+02 | 11 |
| Iamgold | 25.84 | 13 | 2.67E+02 | 14 |
| Kinross | 578.5 | 5 | 1.38E+03 | 5 |
| Lundin Mining | 233 | 9 | 1.24E+03 | 7 |
| New Gold | 141.4 | 11 | 4.66E+02 | 12 |
| Newmont | 1011 | 1 | 1.72E+03 | 3 |
| Teck Resources | 832 | 3 | 1.25E+03 | 6 |

Table 2: Ranking of 15 companies based on $R_{0.95}(0.01)$ and $CVR_{0.95}(0.01)$ measures for a 1-day rainfall event, obtained using the NOAA-CIRES 20th Century reanalysis dataset and mine valuation obtained from broker reports from TD Securities

| Company | $R_{0.95}(0.01)$ | Rank $R_{0.95}(0.01)$ | $CVR_{0.95}(0.01)$ | Rank $CVR_{0.95}(0.01)$ |
|---|---|---|---|---|
| Agnico Eagle | 5.67E-02 | 10 | 1.68E-01 | 12 |
| B2Gold | 8.95E-02 | 3 | 1.81E-01 | 9 |
| Barrick Gold | 6.43E-02 | 9 | 1.73E-01 | 11 |
| Capstone Mining | 0.00E+00 | 14.5 | 1.84E-01 | 7 |
| Eldorado | 8.95E-02 | 4 | 1.73E-01 | 10 |
| First Quantum Mineral | 1.43E-02 | 13 | 3.81E-01 | 1 |
| Franco Nevada | 7.54E-02 | 6 | 1.28E-01 | 15 |
| Goldcorp | 6.91E-02 | 7 | 1.47E-01 | 13 |
| Hudbay | 0.00E+00 | 14.5 | 2.09E-01 | 6 |
| Iamgold | 2.22E-02 | 12 | 2.29E-01 | 4 |
| Kinross | 9.41E-02 | 2 | 2.25E-01 | 5 |
| Lundin Mining | 6.54E-02 | 8 | 3.48E-01 | 2 |
| New Gold | 5.55E-02 | 11 | 1.83E-01 | 8 |
| Newmont | 8.56E-02 | 5 | 1.45E-01 | 14 |
| Teck Resources | 0.1696228 | 1 | 0.2545084 | 3 |




A first thing that can be noted is that in our examples, ranks vary significantly depending on the use of the $S_t'(p)$ or $R_t(p)$. For instance, Barrick Gold and Newmont Corporation both appear amongst the most exposed companies (for both measures) when using $S_t'(p)$, but much less when using $R_t(p)$, which makes sense as these two companies are rather large, but have a relatively diverse portfolio in terms of their geographical locations and climate exposure. A second observation is that the two indexes

do yield some differences in ranking, in particular for First Quantum Mineral. This company has relatively few mines (9 assets are valued in the broker report), with a relatively important geographical variability; some of these assets are small, while a couple of projects are fairly large and with very high valuation. This explains the discrepancy between the two indices. Finally, for cases in which the number of hits recorded is too low overall (e.g. Hudbay), our indexes (and in particular the quantile one), might not be usable when working with empirical data.

**6. Summary and Discussion**

Global water risk including scarcity, flooding, pollution and anthropogenic climate change is of increasing concern to investors, companies, regulators and governments worldwide. Despite the recognition that these factors exist, an approach towards portfolio risk assessment that accounts for the geographical distribution of assets in a portfolio, and the associated exposure to climate extremes has not emerged. Such an assessment is of growing interest in particular to long term investors

who are the owners of these multi-national businesses and currently lack a concrete methodology to compare the relative risks associated with different companies comprising their investment portfolios.

This paper represents perhaps the first effort to address this gap. A simple index that can obtained through weighting by appropriate financial measures of exposure was developed and illustrated. Our hypothesis was that businesses with agricultural supply chains, and the mining industry, were likely to have significant spatio-temporal correlation in their asset level exposure

that could potentially lead to a fat tailed exposure at the portfolio level. The mining industry presented an opportunity for exploring such risks, given that the locations of mines, and various attributes related to the mines can be readily ascertained from publicly available information. Further, as engineered enterprises, it is common for mining companies to use risk based design criteria for structures intended to mitigate the impact of extreme rainfall related hazards at each mine.

While the short climate records typically used to estimate the design parameters for such structures translate into considerable

uncertainty as to the appropriate level of design, the fact that a structure is being designed with a nominal annual probability of failure p directly translates into an estimate of the residual risk that the enterprise is exposed to. Consequently, if long climate records or projections are available, then one can estimate how the residual risk or exposure at each site varies with time, and also if multiple such events could happen in the same year across a portfolio of mines. This observation opens up the possibility of exploring the spatial and temporal clustering of risk exposure and its manifestation at the portfolio level,

whether the portfolio is composed of multiple companies, or a single company; is concentrated in a particular sector of mining, e.g., copper, or is diversified; and whether it is largely based in one country or is geographically diversified. An investor or a company can then seek to understand and mitigate the portfolio risk through appropriate hedging mechanisms.



Since at site climate records are usually short, and it is difficult to pull together global coverage, we considered the use of global, gridded daily rainfall estimated by two different Climate re-analysis models, from NCAR with a 164 year record, and from the ECMWF with a 111 year record. These models embody the same physics of ocean-atmosphere circulation that is used in the models used for seasonal climate forecasts, or for the projection of future climates. However, they are run over a

long historical period and are "corrected" daily over that period to best match the observed surface temperature and pressure data for each historical day. Since the number of observations available varies over the historical period, and the models have different spatial resolution and correction schemes, their retrospective projections do not always agree. Rather, just as the IPCC models for future projections represent an ensemble of possibilities, so do the retrospective or re-analysis simulations. We expect that the large scale features and teleconnections in these models will be similar, but the precise magnitudes of events at

specific locations on specific days and years will not match. Consequently, our approach considers the yearly number of events that exceed a specified quantile computed internally for that location for each such model. In other work, this quantile based approach has been recognized as effective at addressing the biases in each individual model's projections relative to observations.

Our investigations of selected, representative mining portfolios demonstrate that there is significant spatial and temporal

clustering in the exposure of mines to the 2 criteria we considered, a 100 year 1 day annual maximum rainfall event, and a 10 year 30 day annual maximum rainfall event. In some cases, for the worst year in the 164 year record, the total number of exceedances of a 10 year event, i.e., with a yearly probability of occurrence of 0.1 of the residual risk at any given site, was very close to the total number of mines under consideration. This happens because there are several independent events in that year at multiple sites that exceed the design threshold. The consequence is that the portfolio exposure in this setting is much,

much greater than the nominal $pm$ (i.e. the probability of exceedance at each mine multiplied by the number of mines). There is also evidence that the frequency of exceedance of such events at the scale of the mine portfolio is increasing over time, and that pronounced decadal variations in this exposure risk are notable for all 4 companies analysed.

In this paper, we considered two financial metrics for weighting the exposure to the residual risk at each site. These were an estimation of the revenue generated by each mine (calculated as production multiplied by the commodity price) for the most

recent year, and the net asset value estimated for each mine in a recent valuation completed by mining financial research analysts. The exposure of the portfolio rather than a single mine is of interest, in particular to indicate to an investor the potential relative impact of a temporary or permanent production disruption as a result of the risks discussed above on a given company's financial performance. Since the likely loss at each mine if the design event is exceeded is hard to estimate a priori, even by the mining company, one needs an approach that allows an appropriate weighting of the potential portfolio losses. We

intend for the index we developed to be used for sensitivity analysis, to explore how the total portfolio exposure may scale depending on various levels of designed risk protection. Consequently, we assumed that the loss at a certain level of design (average annual probability of exceedance of the annual maximum rainfall) is proportional to either the revenue at each site, or to the net asset value at each site, for each event that exceeds the design level. To estimate the potential impact of temporary production disruptions that a given mine may experience relatively frequently (e.g., with a 10 year return period), we use an



approximation of the revenue by mine as a proxy to weight the number of events of that magnitude experience at each site. For more catastrophic events, e.g., those related to the 100 year annual maximum rainfall that may result in permanent mine closure (or a full write-off of a given asset), we weight the frequency of such events in each year by the net asset value of each mine, providing a measure of the portfolio exposure. By applying these weights, we discovered that the portfolio financial

exposure:

- typically increases over time, with decadal variations, as expected given the space and time clustering of the frequency of exceedances

- the tail of the probability distribution of portfolio risk for different companies may behave very differently;

These observations reflect geographical aspects of the structure of portfolio risk, and could motivate a company to hedge such

risks using parametric or index insurance mechanisms or other financial risk management instruments. For an investor a characterization of the geographical nature of risk, as well as that of portfolio risk can permit risk balancing strategies through an appropriate weighting of companies, sectors or geographies.

Furthermore, this same methodology could be employed at the investor portfolio level rather than at the company level. Investors often own a collection of companies, each with a subset of assets which are inherently exposed to their own subsets

of risks.  Investor portfolios could be disaggregated into their individual components (asset by asset) and different portfolio constructs could be assigned different risks based on their exposure to extreme rain events. Rebalancing exercises could consider effects over both space and time to the risks considered in this paper.

A key question that emerges is whether these climate risk factors actually translate into significant financial risks relative to other financial risk factors associated with investments in mining or other multi-national enterprises. The answer to this

question requires a disclosure from mining companies of their design processes, the associated residual risk and estimate of the loss incurred if a failure event occurs. A first part of this process is an internal assessment of these factors by mining companies, and hence a first order impact of our paper could be a self-examination of these issues by mining companies, and the use of the resulting information to re-evaluate their risk management processes. We know that losses from some such events can be significant. Reported mining related losses from the extreme rainfall event in the Atacama Desert in Chile in

March 2015 were estimated to be of the order of $1 billion insured, and a like amount uninsured. This compares with the $1.6 billion in capital expenditures associated with the desalination and pumping project for mining in the same region, which attracted significant attention as an example of water risk. In the absence of more detailed disclosure and internal assessment by mining companies, the best we can do is provide relative rankings of the financial exposure of different companies having distinct portfolios.  Logically we expect extreme rainfall events that result in catastrophic loss will impact a given company's

financial performance, however the analysis we perform in this paper is solely theoretical and on a relative basis.

Social conflict is often cited as the most significant water related risk for mining companies. In our analysis here, we considered a term in Eq. 1 for impacts external to the mine, that the mine owner would be liable for (and hence would directly impact that company's financial performance). However, in the examples in this paper we did not develop estimates for the potential liabilities that would come from the ecological, environmental and social impacts downstream of the failure of mine





infrastructure. This is an area where we plan to make further headway, in collaboration with WWF Norway, who has developed a database that maps mines, ecosystems and human habitations that are interconnected by the natural drainage network, and hence are the potential for direct impact if mine systems are overcome by extreme rainfall.

A second area of social conflict related to water emerges not from pollution or flooding, but from water scarcity. For an existing mine, this is manifest during a severe, sustained drought. In this setting, even existing senior water rights or water access arrangements can be strained. The basic idea of residual risk for climate extremes that we introduced in this paper can also be extended to the drought case, with the proviso that a quantification of the competition for water under these conditions that would be faced by the mining company, and an assessment of their plans to deal with such contingencies would be required. While some generalized products (e.g., (WRI, 2015)) claim to provide estimates for such water risks, we believe that mining companies need to assess these risks internally relative to different severity and durations possible for droughts; integrate the analyses into their risk management processes and provide disclosure of these risks at a site by site level. This site level analysis could then be aggregated to determine the financial impact from the investor and regulator perspectives.

We noted earlier that climate information is marked by uncertainty and structured space-time variability. We were able to tap a few realizations of such variability using two climate re-analysis products. However, many more such products are available from 1948 to now, 1979 to now, and 1997 to now. For drought there are also global reconstructions of paleo-drought from tree rings and other proxies, which provide a window into climate variability over the last 5 centuries or more. The spatial resolution of climate information, as well as the fidelity to ground observations varies. Similarly, ground based observations of varying duration are available. It is indeed possible to build nonstationary, stochastic simulation models that integrate across such sources of information and provide simulations that can be used to reduce the uncertainty associated with the risk of climate extremes that may vary across space and time. Our past work has addressed some of these issues, and we expect that the tools developed for those cases can also be applied here. However, a bigger issue that needs to be addressed is the estimation of potential financial loss and the attendant uncertainty covering both impacts internal and external to the mine.

To facilitate climate informed portfolio risk analyses, we have developed a Web-based App, using the R statistical platform, that can accept the location of multiple sites – mines or other assets in a portfolio; the specification of the duration and rarity of the rainfall extremes of interest; estimates of the financial exposure at each site; and other parameters from a user, and allow them to compute the portfolio risk measures presented here. A selection of the climate data that are available to use is also available.

**Data availability**

Climate data can be found at the following links:

NOAA-CIRES 20th Century V2C daily precipitation data: http://www.esrl.noaa.gov/psd/cgi-bin/db_search/DBSearch.pl?Dataset=NOAA-CIRES+20th+Century+Reanalysis+Version+2c&Variable=Precipitation+Rate





ECMWF-ERA 20C: http://rda.ucar.edu/datasets/ds626.0/index.html#cgi-bin/datasets/getSubset?dsnum=626.0&action=customizeGrML&_da=y&so=RgpNO&gindex=15

Broker reports are available on the Thomson Reuters platform

**Appendices**

**Appendix A: Portfolio description**

In this appendix, a description of each of the mine portfolio studied is presented. For each portfolio, asset locations are provided. Estimated revenues introduced in 4.2.1 are also shown here for Barrick Gold and Newmont Corporation. However, NAV values introduced in 4.2.2 are not provided, as they came from a non-public data source.

BHP Billiton 2015 mining assets (BHP Billiton, 2016)

Information on the BHP Billiton mining portfolio was obtained by cross-referencing the mining assets mentioned in (BHP Billiton, 2016) and the coordinates from an internet search. It comprised 38 mine sites. It is important to note groups of mines such as the Hammersley system in Pilbara were disaggregated. The orebodies mines in particular were each considered as a given asset. However, due to the difficulty of finding information, they were all assigned the coordinates approximate coordinates corresponding to the Hammersley joint venture.

**Table A1: 2015 mining asset coordinates for BHP Billiton**

| Asset Name | Latitude | Longitude | Primary Commodity | Year | Ownership |
|---|---|---|---|---|---|
| Goonyella Riverside | -21.80889 | 147.97861 | Coal | 2015 | 50% |
| Broadmeadow | -21.8049 | 147.9845 | Coal | 2015 | 50% |
| Daunia | -22.05892 | 148.29836 | Coal | 2015 | 50% |
| Caval Ridge | -22.14199 | 148.06098 | Coal | 2015 | 50% |
| Peak Downs | -22.254 | 148.196 | Coal | 2015 | 50% |
| Saraji | -22.36944 | 148.29111 | Coal | 2015 | 50% |
| Blackwater | -23.68556 | 148.8075 | Coal | 2015 | 50% |
| Norwich Park | -22.61583 | 148.42944 | Coal | 2015 | 50% |



| | | | | | |
|---|---|---|---|---|---|
| Gregory | -23.17222 | 148.35639 | Coal | 2015 | 50% |
| Crinum | -23.17222 | 148.35639 | Coal | 2015 | 50% |
| South Walker Creek | -21.78457 | 148.47162 | Coal | 2015 | 80% |
| Poitrel | -22.04111 | 148.23444 | Coal | 2015 | 80% |
| Mt Arthur | -32.34833 | 150.90556 | Coal | 2015 | 100% |
| San Juan | 36.80151 | -108.43064 | Coal | 2015 | 100% |
| Cerrejon | 11.018 | -72.714 | Coal | 2015 | 33% |
| Antamina | -9.53917 | -77.05 | Copper | 2015 | 34% |
| Escondida | -24.26889 | -69.07466 | Copper | 2015 | 58% |
| Olympic Dam | -30.44 | 136.88889 | Copper | 2015 | 100% |
| Pampa Norte Cerro Colorado | -24.26667 | -69.06667 | Copper | 2015 | 100% |
| Pampa Norte Spence | -24.26667 | -69.06667 | Copper | 2015 | 100% |
| Cliffs | -27.31306 | 120.55306 | Nickel | 2015 | 100% |
| Leinster | -27.81424 | 120.70243 | Nickel | 2015 | 100% |
| Mt Keith | -27.23056 | 120.545 | Nickel | 2015 | 100% |
| Mt Whaleback | -23.36536 | 119.6754 | Iron Ore | 2015 | 85% |
| Orebody 18 | -23.386157 | 119.988638 | Iron Ore | 2015 | 85% |
| Orebody 23 | -23.386157 | 119.988638 | Iron Ore | 2015 | 85% |
| Orebody 24 | -23.386157 | 119.988638 | Iron Ore | 2015 | 85% |
| Orebody 25 | -23.386157 | 119.988638 | Iron Ore | 2015 | 85% |
| Orebody 29 | -23.386157 | 119.988638 | Iron Ore | 2015 | 85% |
| Orebody 30 | -23.386157 | 119.988638 | Iron Ore | 2015 | 85% |



| | | | | | |
|---|---|---|---|---|---|
| Orebody 35 | -23.386157 | 119.988638 | Iron Ore | 2015 | 85% |
| Yandi | -22.71889 | 119.06611 | Iron Ore | 2015 | 85% |
| Jimblebar | -23.38083 | 120.13806 | Iron Ore | 2015 | 85% |
| Wheelarra | -23.38145 | 120.13146 | Iron Ore | 2015 | 51% |
| Area C | -22.92362 | 118.97679 | Iron Ore | 2015 | 85% |
| Yarrie | -20.417278 | 120.0100995 | Iron Ore | 2015 | 85% |
| Nimingarra | -20.417278 | 120.0100995 | Iron Ore | 2015 | 85% |
| Samarco | -20.16149 | -43.50515 | Iron Ore | 2015 | 50% |

As can be seen from the map below, there is a high clustering in two regions of Australia: Pilbara and North East Queensland which are important respectively iron and coal producing areas.



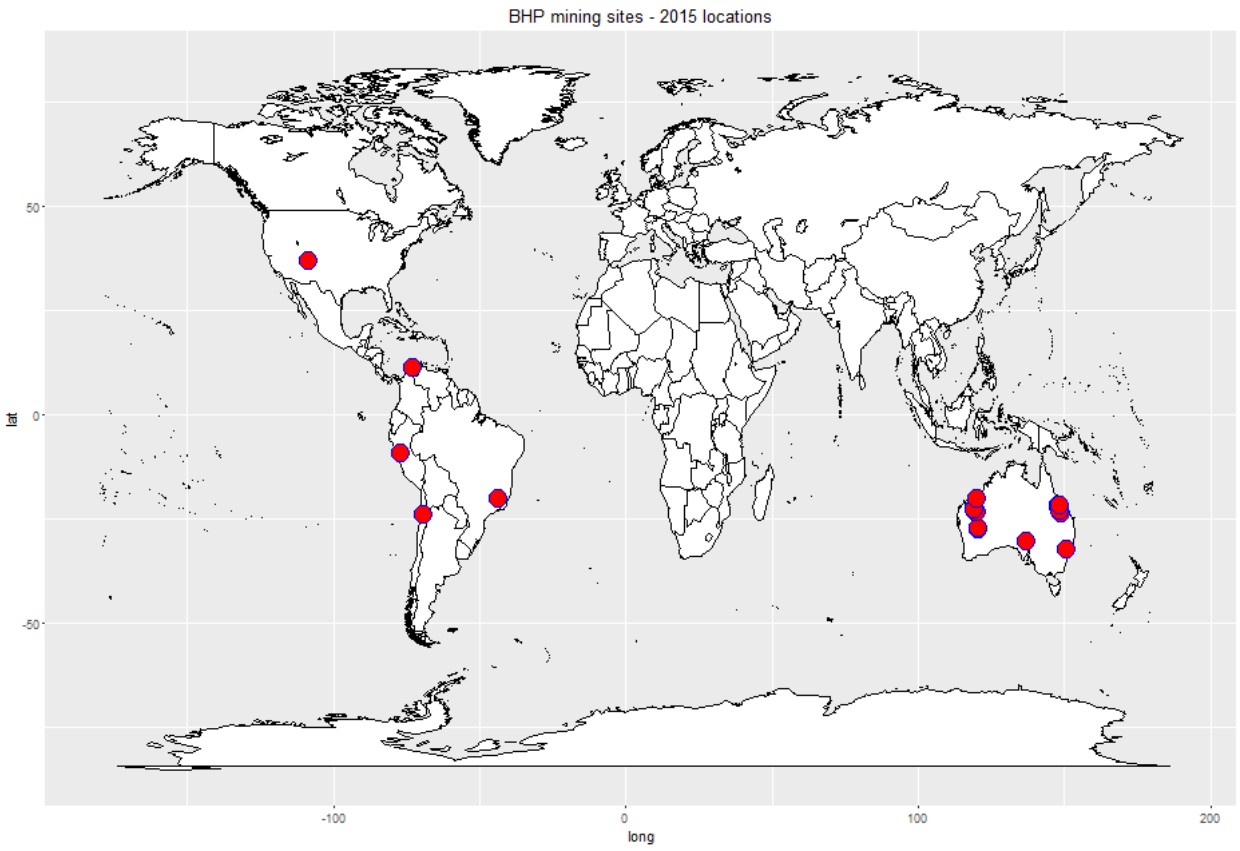

**Fig. A1: Map of BHP Billiton 2015 mining assets**

Barrick Gold 2015 producing assets (Barrick Gold Corporation, 2016)

Information on the Barrick Gold mining portfolio was obtained by cross-referencing the production information mentioned in

5    (Barrick Gold Corporation, 2016) and the coordinates from an internet search. It comprised 19 gold and copper mines.
Commodity price information was retrieved from (Word Bank, 2016).

**Table A2: Mining asset coordinates and attributable revenue per mine for Barrick gold for 2015**

| Asset Name | Primary Commodity | Commodity | Latitude | Longitude | Ownership | Attributable Production | Unit |
|---|---|---|---|---|---|---|---|
| Bald Mountain | Gold | Gold | 39.94139 | -115.543 | 100% | 1.91E+05 | ounces |
| Bulyanhulu | Gold | Gold | -3.22344 | 32.48616 | 64% | 2.74E+05 | ounces |
| Buzwagi | Gold | Gold | -3.861 | 32.67 | 64% | 1.71E+05 | ounces |
| Cortez | Gold | Gold | 40.16973 | -116.608 | 100% | 9.99E+05 | ounces |
| Cowal | Gold | Gold | -33.6374 | 147.4053 | 100% | 1.56E+05 | ounces |





| | | | | | | | |
|---|---|---|---|---|---|---|---|
| Golden Sunlight | Gold | Gold | 45.90578 | -112.022 | 100% | 6.80E+04 | ounces |
| Goldstrike | Gold | Gold | 40.98072 | -116.381 | 100% | 1.05E+06 | ounces |
| Hemlo | Gold | Gold | 48.69755 | -85.9252 | 100% | 2.19E+05 | ounces |
| Jabal Sayid | Copper | Copper | 23.85226 | 40.94042 | 100% | 6.00E+06 | pounds |
| Kalgoorlie | Gold | Gold | -30.553 | 121.45 | 50% | 3.20E+05 | ounces |
| Lagunas Norte | Gold | Gold | -7.94806 | -78.2447 | 100% | 5.60E+05 | ounces |
| Lumwana | Copper | Copper | -12.2362 | 25.82228 | 100% | 2.87E+08 | pounds |
| North Mara | Gold | Gold | -1.47333 | 34.51639 | 64% | 2.87E+05 | ounces |
| Pierina | Gold | Gold | -9.44694 | -77.5869 | 100% | 5.40E+04 | ounces |
| Porgera | Gold | Gold | -5.465 | 143.095 | 48% | 4.36E+05 | ounces |
| Pueblo Viejo | Gold | Gold | 18.93861 | -70.1739 | 60% | 5.72E+05 | ounces |
| Round Mountain | Gold | Gold | 38.70389 | -117.077 | 50% | 1.92E+05 | ounces |
| Ruby Hill | Gold | Gold | 39.52722 | -115.987 | 100% | 1.00E+04 | ounces |
| Turquoise Ridge | Gold | Gold | 41.21639 | -117.256 | 75% | 2.17E+05 | ounces |
| Veladero | Gold | Gold | -29.3714 | -69.9528 | 100% | 6.02E+05 | ounces |
| Zaldivar | Copper | Copper | -24.2186 | -69.0678 | 100% | 2.18E+08 | pounds |

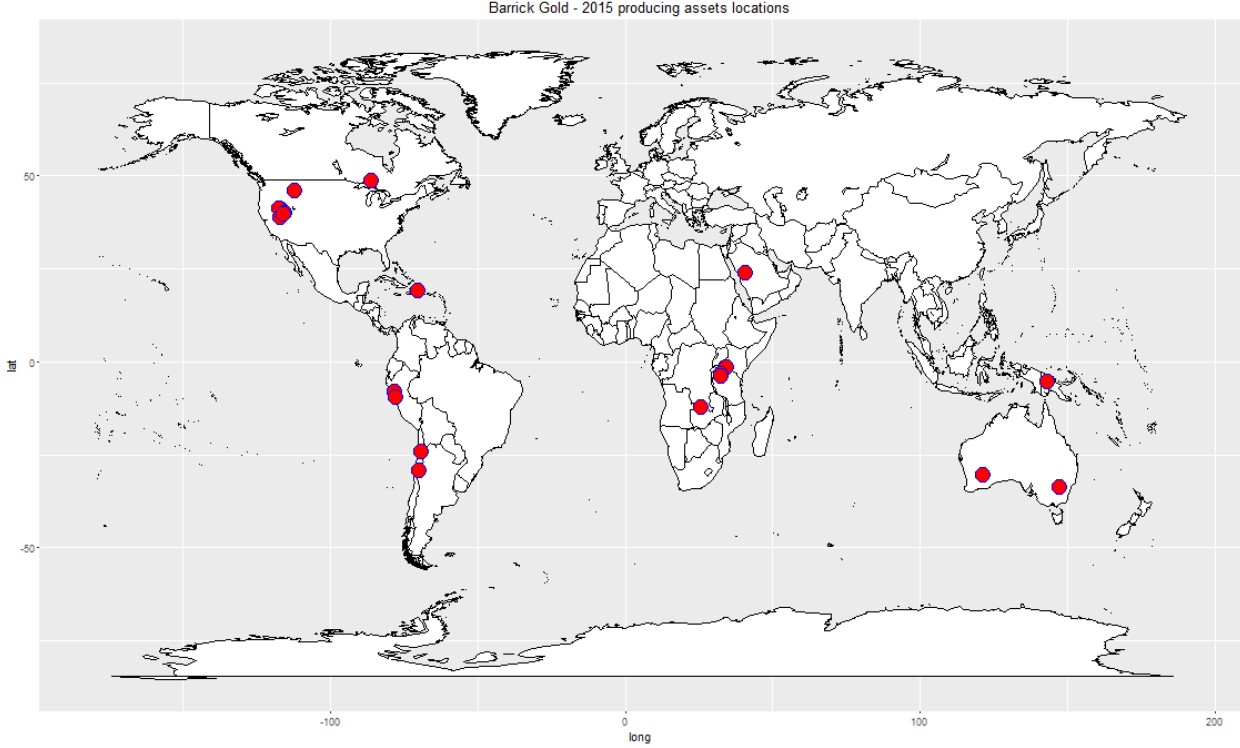

**Fig. A2: Map of Barrick Gold 2015 producing assets**

Newmont 2015 producing assets (Newmont Mining Corporation, 2016)





Information on the Newmont Corporation mining portfolio was obtained by cross-referencing the production information mentioned in (Newmont Mining Corporation, 2016) and the coordinates from an internet search. It comprised 16 gold and copper mines. Commodity price information was retrieved from (Word Bank, 2016).

**Table A3: Mining asset coordinates and attributable production per mine for Newmont Corporation for 2015**

| Asset Name | Primary Commodity | Commodity | Latitude | Longitude | Ownership | Attributable Production | Unit |
|---|---|---|---|---|---|---|---|
| Ahafo | Gold | Gold | 7.03076 | -2.35953 | 100% | 3.32E+05 | ounces |
| Akyem | Gold | Gold | 6.35876 | -1.02607 | 100% | 4.73E+05 | ounces |
| Batu Hijau | Gold | Gold | -8.96667 | 116.8667 | 48.50% | 3.28E+05 | ounces |
| Batu Hijau | Gold | Copper | -8.96667 | 116.8667 | 48.50% | 2.40E+08 | pounds |
| Boddington | Gold | Gold | -32.7417 | 116.3469 | 100% | 7.94E+05 | ounces |
| Boddington | Gold | Copper | -32.7417 | 116.3469 | 100% | 7.90E+07 | pounds |
| Carlin | Gold | Gold | 40.4651 | -117.102 | 100% | 8.86E+05 | ounces |
| CC & V | Gold | Gold | 38.72387 | -105.153 | 100% | 8.10E+04 | ounces |
| Duketon | Gold | Gold | -27.642 | 122.044 | 19.45% | 5.70E+04 | ounces |
| Kalgoorlie | Gold | Gold | -30.7781 | 121.505 | 50% | 3.16E+05 | ounces |
| La Zanja | Gold | Gold | -6.82902 | -78.8941 | 47% | 6.60E+04 | ounces |
| Phoenix | Gold | Gold | 40.53917 | -117.122 | 100% | 2.05E+05 | ounces |
| Phoenix | Gold | Copper | 40.53917 | -117.122 | 100% | 4.60E+07 | pounds |
| Tanami | Gold | Gold | -19.9769 | 129.7139 | 100% | 4.36E+05 | ounces |
| Turquoise Ridge | Gold | Gold | 41.21639 | -117.256 | 25% | 6.80E+04 | ounces |
| Twin Creeks | Gold | Gold | 41.25833 | -117.169 | 100% | 4.03E+05 | ounces |
| Waihi | Gold | Gold | -37.393 | 175.838 | 100% | 1.19E+05 | ounces |
| Yanacocha | Gold | Gold | -6.99417 | -78.5319 | 51.35% | 4.71E+05 | ounces |



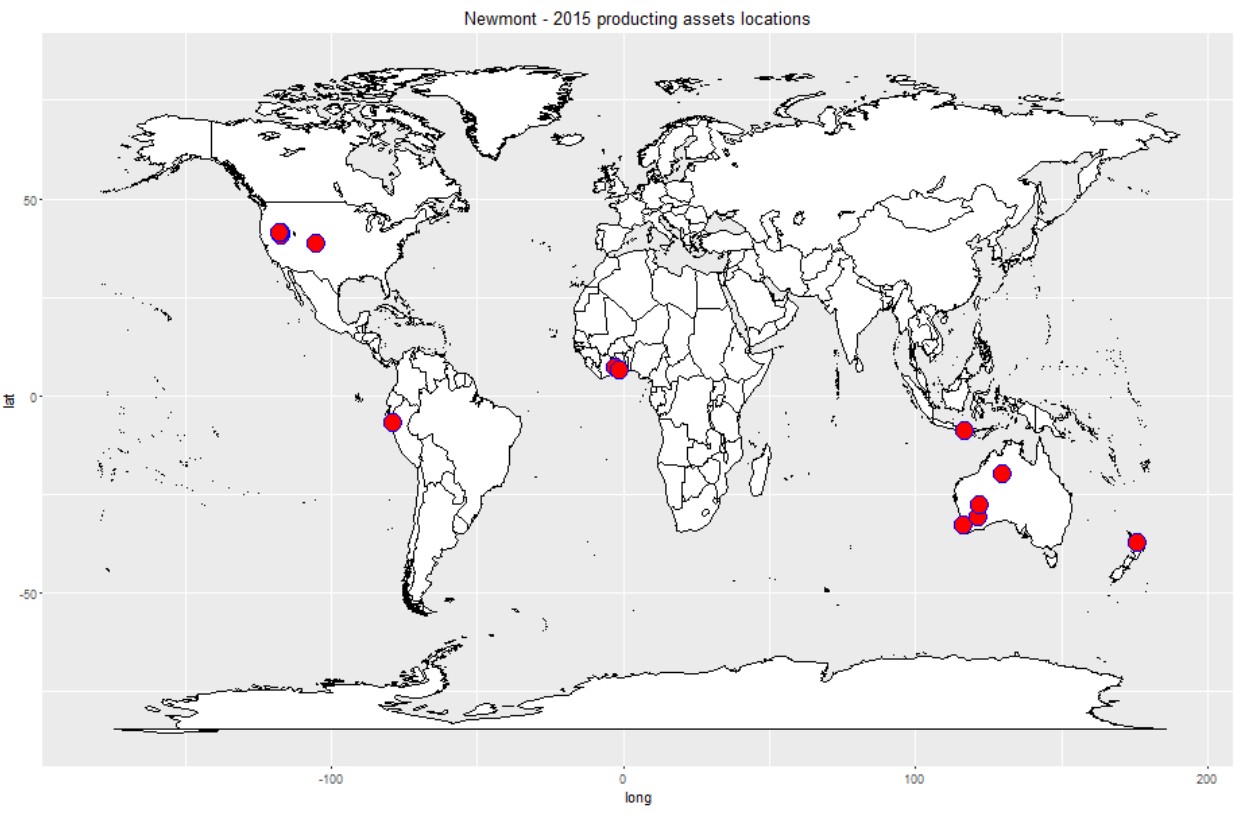

**Fig. A3: Map of Newmont 2015 producing assets**

Rio Tinto 2015 mining assets (Rio Tinto, 2016)

Information on the Rio Tinto mining portfolio was obtained by cross-referencing the mining assets mentioned in (Rio Tinto, 2016) and the coordinates found through an internet search. It comprised 40 mine sites.

**Table A4: 2015 mining asset coordinates for Rio Tinto**

| Asset Name | Latitude | Longitude | Primary Commodity | Year | Ownership |
|---|---|---|---|---|---|
| Gove | -12.295 | 136.83 | Bauxite | 2015 | 100% |
| Porto Trombetas | -1.4717486 | -56.3784885 | Bauxite | 2015 | 12% |
| Sangaredi | 11.1 | -13.77 | Bauxite | 2015 | 23% |
| Weipa | -12.533 | 141.833 | Bauxite | 2015 | 100% |





| | | | | | |
|---|---|---|---|---|---|
| Boron | 35.0331722 | -117.668687 | Borates | 2015 | 100% |
| Bengalla | -32.26667 | 150.85 | Coal | 2015 | 32% |
| Hail Creek | -21.5 | 148.4 | Coal | 2015 | 82% |
| Hunter Valley | -32.525 | 150.98333 | Coal | 2015 | 80% |
| Kestrel | -23.23333 | 148.36667 | Coal | 2015 | 80% |
| Mt Thorley | -32.64726 | 151.07113 | Coal | 2015 | 64% |
| Warkworth | -32.60694 | 151.09028 | Coal | 2015 | 45% |
| Zululand Anthracite Colliery | -28.1598 | 31.6875 | Coal | 2015 | 74% |
| Bingham Canyon | 40.52056 | -112.145 | Copper | 2015 | 100% |
| Escondida | -24.26889 | -69.07466 | Copper | 2015 | 30% |
| Grasberg | -4.05667 | 137.11361 | Copper | 2015 | 40% |
| Oyu Tolgoi | 43.767127 | 107.4462891 | Copper | 2015 | 34% |
| Argyle | -16.73056 | 128.38389 | Diamonds | 2015 | 100% |
| Diavik | 64.49643 | -110.27715 | Diamonds | 2015 | 60% |
| Brockman 2 | -22.59717 | 117.21776 | Iron Ore | 2015 | 100% |
| Brockman 4 | -22.59717 | 117.21776 | Iron Ore | 2015 | 100% |
| Marandoo | -22.63806 | 118.13889 | Iron Ore | 2015 | 100% |
| Mt Tom Price | -22.76821 | 117.76625 | Iron Ore | 2015 | 100% |
| Nammuldi | -22.41222 | 117.3375 | Iron Ore | 2015 | 100% |
| Paraburdoo | -23.22917 | 117.57889 | Iron Ore | 2015 | 100% |
| Western Turner Syncline | -22.66272 | 117.59022 | Iron Ore | 2015 | 100% |
| Yandicoogina | -22.76389 | 119.225 | Iron Ore | 2015 | 100% |





| | | | | | |
|---|---|---|---|---|---|
| Channar | -23.30167 | 117.78889 | Iron Ore | 2015 | 60% |
| Eastern Range | -23.24389 | 117.65694 | Iron Ore | 2015 | 54% |
| Hope Downs 1 | -22.94667 | 119.12306 | Iron Ore | 2015 | 50% |
| Hope Downs 4 | -23.14583 | 119.57889 | Iron Ore | 2015 | 50% |
| IOC | 53.04112 | -66.94422 | Iron Ore | 2015 | 59% |
| Mesa A | -21.68052 | 115.88057 | Iron Ore | 2015 | 53% |
| Mesa J | -21.75 | 116.24 | Iron Ore | 2015 | 53% |
| West Angelas | -23.19056 | 118.78806 | Iron Ore | 2015 | 53% |
| Dampier | -20.7064 | 116.7425 | Salt | 2015 | 68% |
| QMM | -25.0370535 | 46.9295919 | Titanium | 2015 | 80% |
| RBM | -28.6829452 | 32.1305466 | Titanium | 2015 | 74% |
| RTFT | 50.5457265 | -63.3852768 | Titanium | 2015 | 100% |
| Ranger | -12.6851397 | 132.9092073 | Uranium | 2015 | 68% |
| Rössing SJ | -22.509068 | 15.0356483 | Uranium | 2015 | 69% |



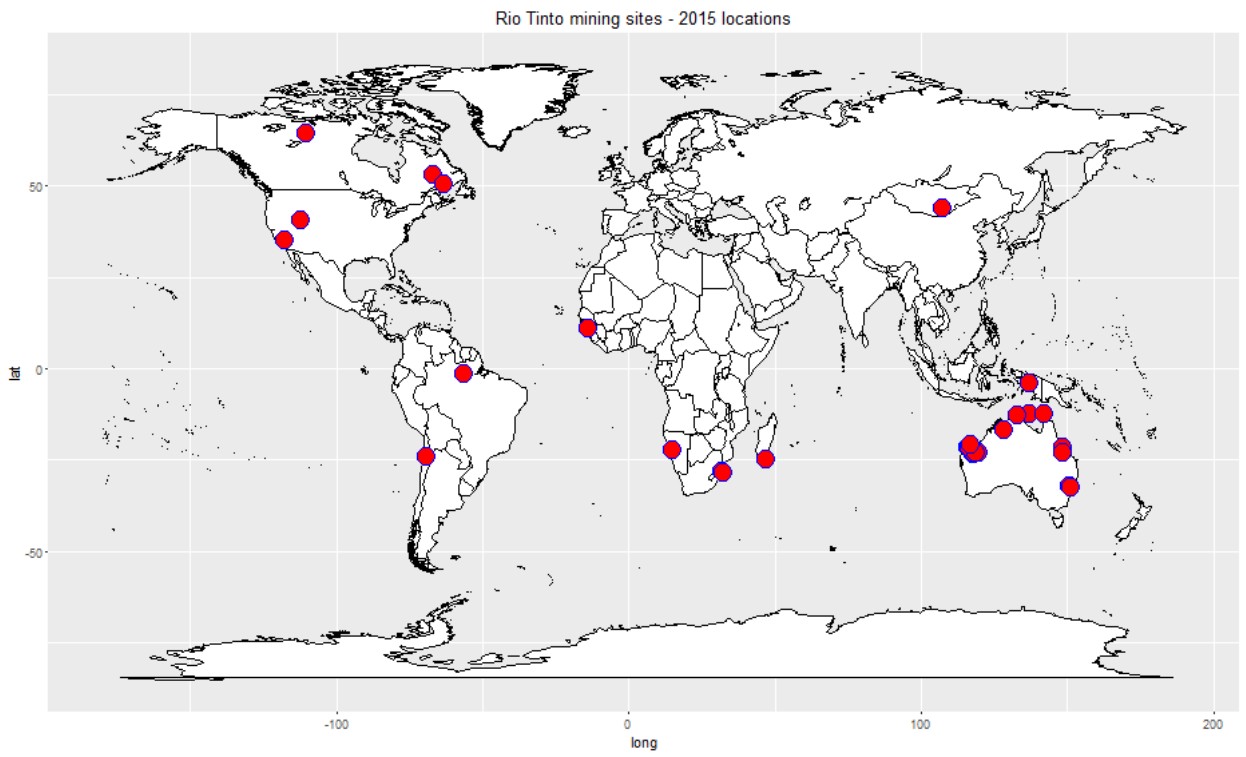

**Fig. A4: Map of Rio Tinto 2015 mining assets**

Barrick Gold mining assets valued in (TD Securities, 2016 a.):

**Table A5: Mining assets valued in (TD Securities, 2016 a.) for Barrick Gold**

| Asset Name | Latitude | Longitude | Ownership |
|---|---|---|---|
| Bulyanhulu | -3.22344 | 32.48616 | 64% |
| Buzwagi | -3.861 | 32.67 | 64% |
| Cerro Casale | 40.16973 | -116.608 | 75% |
| Cortez | -27.7906 | -69.2994 | 100% |
| Donlin Creek | 62.045 | -158.198 | 50% |
| Goldstrike | 40.98072 | -116.381 | 100% |
| Hemlo | 48.69755 | -85.9252 | 100% |





| | | | |
|---|---|---|---|
| Jabal Sayid | 23.85226 | 40.94042 | 50% |
| Kalgoorlie | -30.553 | 121.45 | 50% |
| Lagunas Norte | -7.94806 | -78.2447 | 100% |
| Lumwana | -12.2362 | 25.82228 | 100% |
| North Mara | -1.47333 | 34.51639 | 64% |
| Pascua-Lama | -29.3231 | -70.0233 | 100% |
| Porgera | -5.465 | 143.095 | 48% |
| Pueblo Viejo | 18.93861 | -70.1739 | 60% |
| Turquoise Ridge | 41.21639 | -117.256 | 75% |
| Veladero | -29.3714 | -69.9528 | 100% |
| Zaldivar | -24.2186 | -69.0678 | 50% |
| Other | NA | NA | 100% |

On the following map, asset symbols are proportional to the share of the total NAV they represent:




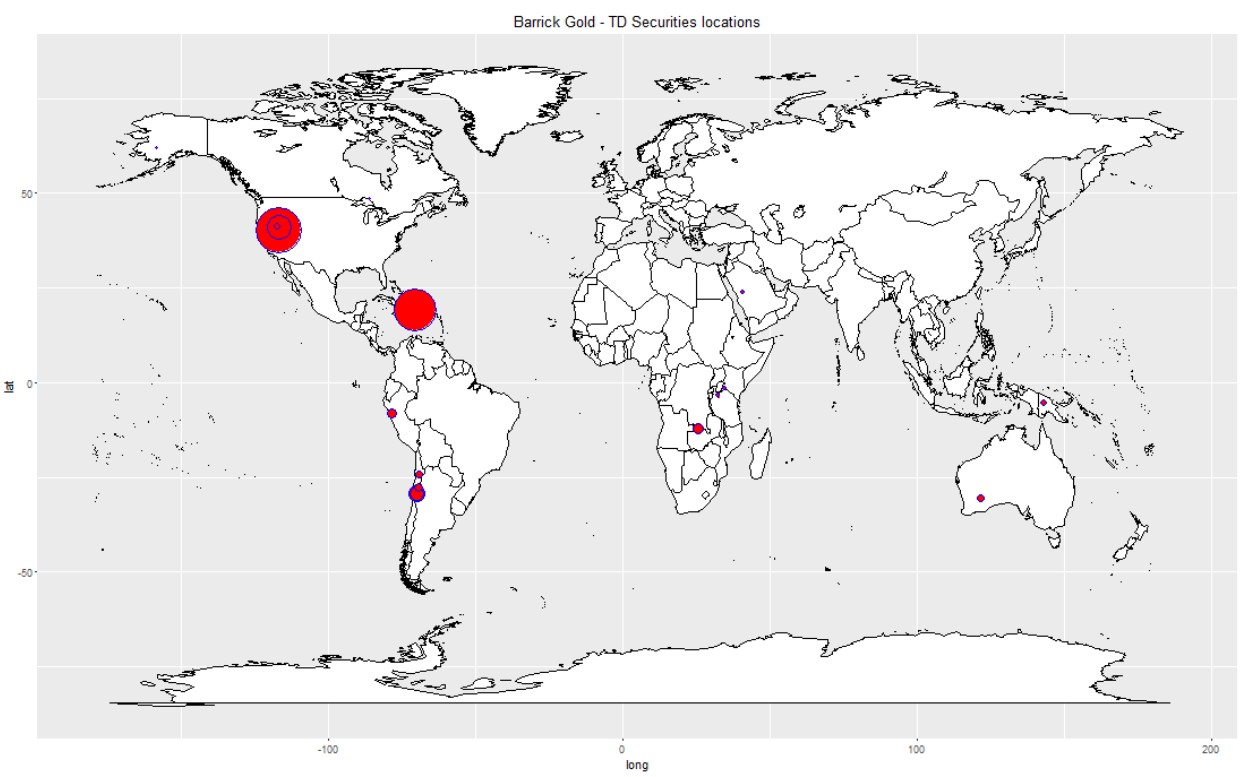

**Fig. A5: Map of Barrick Gold assets reported in** (TD Securities, 2016 a.)

Newmont Corporation mining assets valued in (TD Securities, 2016 b.):

**Table A6: Mining assets valued in (TD Securities, 2016 b.) for Newmont Corporation**

| Asset.Name | Latitude | Longitude | Ownership |
|---|---|---|---|
| Nevada | 40.4651 | -117.102 | 100.00% |
| Cripple Creek & Victor | 38.72387 | -105.153 | 100.00% |
| Yanacocha | -6.99417 | -78.5319 | 51.40% |
| Batu Haijau | -8.96667 | 116.8667 | 44.60% |
| Boddington | -32.7417 | 116.3469 | 100.00% |



| | | | |
|---|---|---|---|
| Kalgoorlie | -30.7781 | 121.505 | 50.00% |
| Tanami | -19.9769 | 129.7139 | 100.00% |
| Ahafo | 7.03076 | -2.35953 | 100.00% |
| Akyem | 6.35876 | -1.02607 | 100.00% |
| Conga M&I | -6.08424 | -78.3616 | 51.40% |
| Merian | 5.125 | -54.5467 | 75.00% |
| Other | NA | NA | 100% |

On the following map, asset symbols are proportional to the share of the total NAV they represent:

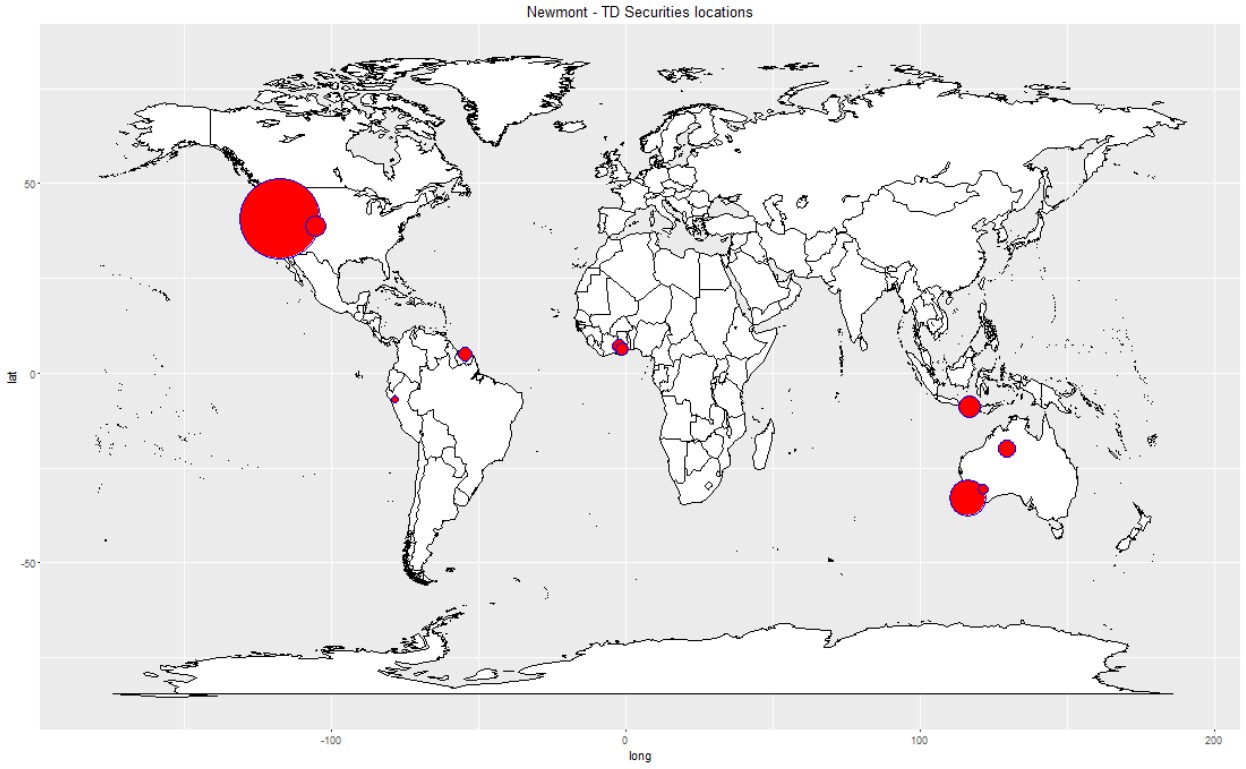

**Fig. A6: Map of Newmont Corporation assets reported in** (TD Securities, 2016 b.)





The similarity between Barrick Gold and Newmont Corporation in terms of the localization of their assets and the value corresponding to given locations is here confirmed.

**Appendix B: Clustering in time and trend**

In this Appendix, we show the analysis of $N_t(p)$ of all the portfolios mentioned in 5.1.1. We consider both a 1-day extreme
5    rainfall event with a 100-year return level and a 30-day extreme rainfall event with a 10-year return level. We also use both
the ECMWF and NOAA reanalysis datasets and therefore restrict the time range to 1900-2010 for consistency. The p-values
from Mann-Kendall tests performed on the time series are indicated above each plot (computations performed for the period
1900-2010).

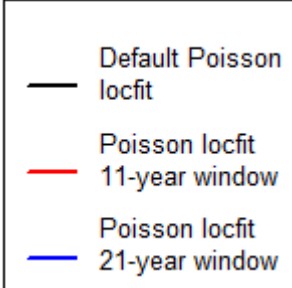













**Figure B1: Time series $N_t(p)$ of the yearly number of 1-day (left) and 30-day (right) extreme rainfall events hitting the four mine portfolios computed from two climate datasets: NOAA-CIRES 20th Century V2c and ECMWF-ERA 20C**

**Table B1: Mann-Kendall test results for the count of the number of extreme 30-day extreme rainfall event with a 10-year return level on four mine portfolios, using both the NOAA and ECMWF climate datasets (computations performed for the period 1851-2014 for the NOAA-CIRES data and 1900-2010 for ECMWF)**

| | NOAA-CIRES V2C | | ECMWF-ERA-20C | |
|---|---|---|---|---|
| Company | Slope sign | Sen slope (p-value) | Slope sign | Sen slope (p-value) |
| Barrick Gold | + | 0.017 (2.9e-10) | | 0 (0.14) |
| BHP Billiton | + | 0.016 (2.9e-10) | | 0 (7.4e-2) |
| Newmont Corporation | + | 0.011 (3.0e-9) | | 0 (0.18) |
| Rio Tinto | + | 0.042 (2.7e-17) | + | 0.019 (9.9e-3) |

**Table B1: Mann-Kendall test results for the count of the number of extreme 1-day extreme rainfall event with a 100-year return level on four mine portfolios, using both the NOAA and ECMWF climate datasets**

| | NOAA-CIRES V2C | | ECMWF-ERA-20C | |
|---|---|---|---|---|
| Company | Slope sign | Sen slope (p-value) | Slope sign | Sen slope (p-value) |
| Barrick Gold | | 0 (1.2e-5) | | 0 (2.0e-2) |
| BHP Billiton | | 0 (2.4e-3) | | 0 (0.65) |
| Newmont Corporation | | 0 (4.4e-5) | | 0 (1.1e-2) |
| Rio Tinto | | 0 (2.8e-9) | | 0 (0.18) |

The main conclusion to be drawn here is that while there seem to exist a positive trend in for the 30-day event many cases, in general, the significance level is lower when using the ECMWF dataset; the p-values on the figure show that, at the 5% level, significant positive trends were detected almost systematically using the NOAA-CIRES 20th Century reanalysis dataset, but that it is only true for certain cases using the ECMWF-ERA 20C data. This emphasizes the need to take a critical approach towards those results, and the value of using multiple reanalysis models.

Furthermore, all the sen slopes for the 100-year event are null, which is due to the fact that sen slopes are computed as the median of the slopes between the points of a given dataset.





## Appendix C: Clustering in space

In this appendix, we show the cdfs corresponding to the analysis of paragraph 4.1.2, considering both a 1-day extreme event and a 30-day extreme event.







**Figure C1: Comparison of the cdfs of the yearly number of 1-day (left) and 30-day (right) extreme rainfall events hitting the BHP, Barrick Gold, Newmont and Rio Tinto mine portfolios with the corresponding theoretical cdfs assuming independence of events in space and time (Poisson processes). Empirical distributions were derived from the $N_t(p)$ time series using the ecdf R function. The NOAA-CIRES V2c dataset was used**

**Figure C2: Comparison of the cdfs of the yearly number of 1-day (left) and 30-day (right) extreme rainfall events hitting the BHP, Barrick Gold, Newmont and Rio Tinto mine portfolios with the corresponding theoretical cdfs assuming independence of events in space and time (Poisson processes). Empirical distributions were derived from the $N_t(p)$ time series using the ecdf R function. The ECMWF-ERA 20C dataset was used.**





As previously evoked, in each case, the empirical distribution differs significantly from the theoretical Poisson process associated, with, in particular, a fatter tail. This is confirmed by the study of the $r_k$ ratios:

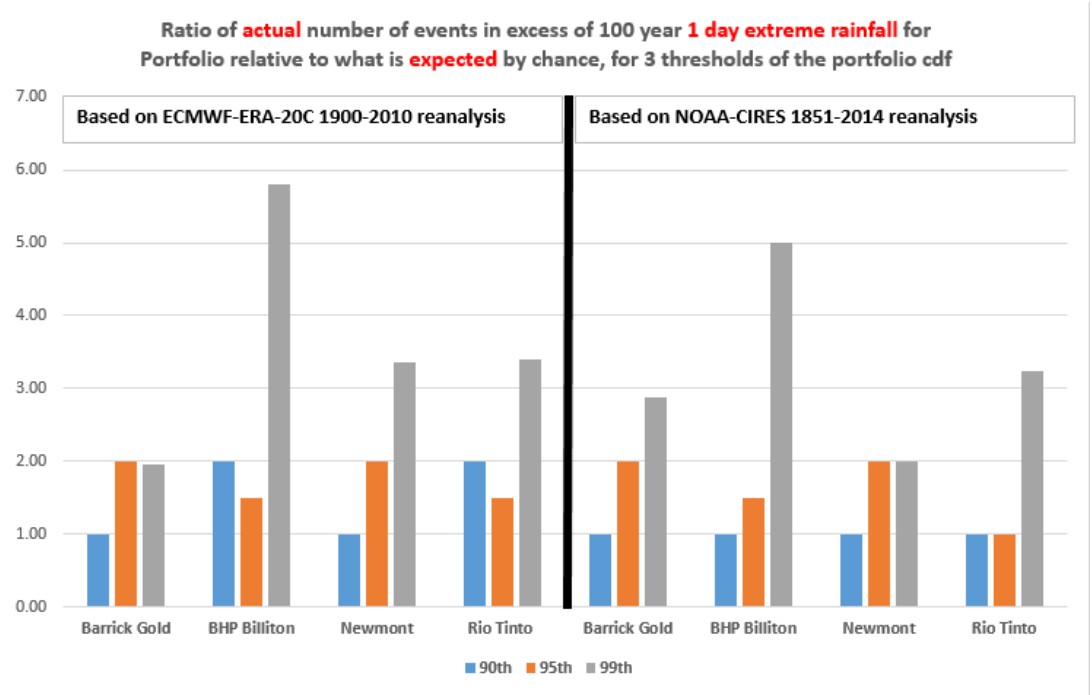

**Figure C3: ratio of the actual number of the number of 10 year 10 day extreme rainfall events fitting the four portfolio relative to what is expected by chance, for 3 thresholds of the portfolio cdf**





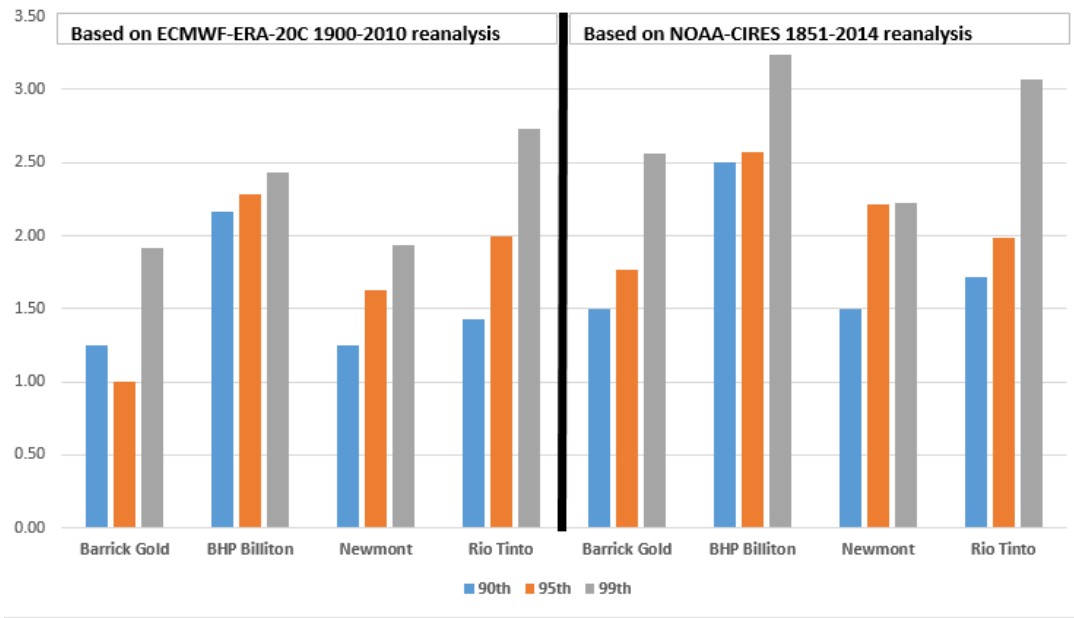

**Figure C4: ratio of the actual number of the number of 10 year 30 day extreme rainfall events fitting the four portfolio relative to what is expected by chance, for 3 thresholds of the portfolio cdf**

While BHP seems to be the portfolio with the most significant tail exposure in terms of number of hits, one should note that

5    the level of disaggregation of the BHP mine groups we decided on implied to consider each of the Orebodies mines (a group of mines close to each other in the Pilbara region) as an individual asset; this may or may not correspond to an investor's perspective. In any case, from an investor perspective, what ultimately counts is the value exposed rather than the number of events across a portfolio.



**Acknowledgements**

The research presented here was supported by a grant from Norges Bank Investment Management. We acknowledge help from

Ajay Vempati for helping compile the NAV values of the mines from broker reports.

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





**List of tables**

**List of figures**