# Peer review of "A water risk index for portfolio exposure to climatic extremes: conceptualization and an application to the mining industry"

_Hydrology and Earth System Sciences, 2016_

## Referee Comment (RC1) · A. Money (Referee) · 28 Nov 2016

The authors have rightly identified the asymmetries that may exist in the time periods that are applied as terms of reference in the operational design level of mining facilities; and the time periods that are salient to long-term institutional investors who have exposure to the mining companies who own and operate those facilities. They have also made interesting insights into the systemic climate risks that might be embedded across the asset bases of their portfolio companies - and the paper makes that point that while these risks may not be known, they are not necessarily unknowable.

The descriptive connections that the authors make between climate variability and financial impact (p 4-5) are reasonable. However given the substantive arguments of

this paper it would have been interesting to see some empirical longitudinal dataset that considers e.g. changes in production/ output at specific mining sites, correlated to observed rainfall, with appropriate adjustments. It would help substantiate the argument that extreme rainfall has material consequences for mining outputs, with the attendant financial consequences this represents. As it stands, that assumption seems intuitively correct, but it is hard to calibrate rainfall with other effects on mining output - which is necessary to calibrate the significance of this work.

In terms of 'exceedance' data and fat tails, the results do make an interesting contribution to the discussion of systemic climate risk. The authors are however obliged to make a number of assumptions in terms of the likely financial consequences associated with 'excess exceedance' of 1 day/ 100 year and 30 day/ 10 year rainfall. These look somewhat generic, in particular assumptions of 'disruption in production' of 12.5% and 'destruction of NAV' of 10% (qualified as "likely a low number"). Applying these against the companies' revenue and asset value numbers generates a significant result - but this aspect of the methodology seems rather unsubstantiated: certainly when compared to the care given to the rainfall modelling aspect.

Overall this paper does much to help focus academic and practitioner interest on asset level impacts, and the preponderance of systemic risk. But in terms of the connections to financial value at risk, the outcomes are probably hard to substantiate. It would help to analyse longitudinal datasets of production values from different mines, correlating this with observed rainfall records. The revenue and NAV of mining companies is also highly subjective to market commodity prices, and some discussion of the salience of price volatility - relative to climate volatility - in terms of financial risk to mining companies would have been interesting to see in the discussion.

---

## Referee Comment (RC2) · Z. Wang (Referee) · 7 Dec 2016

The authors' work is relevant to the emerging socio-hydrology community and meets the scope and focus of the journal. Global water risks that manifest in financial sector remain a critical concern, this article have made lucid but effective efforts to address such challenges from a perspective of Sovereign Wealth Funds on the specific topic of extreme precipitation induced losses for one or a portfolio of mines. Although corresponding hazards, exposures and risks have been well studied for a long time particularly by insurance and reinsurance companies, the water risk at portfolio level implying inherent spatio-temporal correlation (from ENSO e.g.) has never seen quantified for comparison, at least not publicly.

[Figure]

I particularly like 2 points in the article: 1) The methodology of developing a global water risk index from long-term reanalysis datasets. I have seen many studies on the development of global water-related indicators, while how the index developed in this article is insightful (part 2 "structuring a risk index for climate extremes"). It provides a concise view on massive globally-historical data. 2) applying general extreme value distribution to precipitation time series makes the exceedance probability much more realistic than using empirical distribution (P18 – 22, "evidence of very fat tails for the portfolio risk"). I believe this application and fat tails finding have substantial implication for portfolio management. However, I would point out some confused points need to be better clarified and explained to hopefully improve the paper in some respects.

The major damages for a mine as mentioned in part 2 are spills and tailings dam. Precipitation is only one factor among all related climate or hydrological factors leading to those damages e.g. soil texture, runoff generation, groundwater level etc. It is much better but indeed very hard if not impossible to consider many involved factors both from data side and from physical mechanism side. It is also not conclusive to tell which is more informative when selecting one or several best variables. In spite of those, the combination of extreme precipitation and the capacity of runoff generation is usually regarded as appropriate balance of comprehension and feasibility in general situation. Therefore, I would suggest the authors expand the discussion of devising risk index in part 2 from more hydrological perspectives to better explained the reason selecting precipitation only. Integrating more hydrological variables with precipitation could be one improvement in the future.

The relation between "climate exposure" and "financial risk" is the central of the research and is built up on estimation of two variables, i.e. exceedance risk (eq. 1)

$$n_{i,t}^p$$

and potential loss (eq. 3)

$$L_i(p)$$

[Figure]

A basic question raised here is to what extent eq. 1 and corresponding stationary and non-stationary threshold selection would represent "climate exposure"? It probably be more convincing if calibration or(and) validation is included in Example application parts, or mines in Queensland, South Africa introduced in part 1 are used. And same question to eq. 3 with regard to production loss assumptions (12.5%, 11%).

As far as I understand, eq.5 and eq.6 are absolute and relative value of the same concept. So accordingly the difference between Table 1 and Table 2 is a multiplier, the value of the mines those companies hold. Based on above, I do not fully understand para. 1 in page 25 on the analysis of implications from different ranking, e.g. how "geographical variability" or "discrepancy" is associated with different ranking.

In general, I think this article is an interesting and meaningful work having connected water risk and real world loss and the methodology proposed has potential to be practical framework for water risk managers by adding their insider information. Besides, some minor comments and technical corrections:

1) P4, L19: "300 hundred houses" → "300 houses"? please re-check the number;

2) P6, L23: Is "web based application" available? If so, better to add a link in the article;

3) P11, L20: please specify full name of "NAV" at first mention;

4) Fig. 3 – 8: not as sharp as those in Appendices, re-print of high-res probably needed;

---

## Author Comment (AC2) · 3 Jan 2017

Dear Dr. Wang, Thank you very much for your comment. You definitely caught the direction, significance and initial nature of the work that we intended. - Regarding the issue of presenting devising risk index in part 2 from more hydrological perspectives, it is something that we can discuss a little bit. We are currently in the process of gathering data on failed tailings dams to attempt to build our understanding of this issue. Overall, it has been a challenging process, we have been limited by data access in that regard, and this and the fact that overtopping is one of the major reason behind dam failures led us to consider precipitation data. However, we definitely hope to include more hydrological considerations in a later paper. - Regarding the return

level of the Queensland floods, this changes a lot from station to station. For instance, using the high quality data from the Australian Bureau of Meteorology (BMO, 2016): http://www.bom.gov.au/climate/change/hqsites/, the highest 30-day cumulative rainfall of the December 2010-February 2011 period corresponds to an approximately 8.5 year return-level in Barcaldine, a 3-year return level in Macknade and almost a yearly occurrence in Winton. - The production loss assumptions were once again based on discussion we have had with experts rather than empirical analysis due to the lack of data access. For the Queensland floods, we noticed the distribution in the figure below regarding the percent change in production between 2010 and 2011, based on data reported from 38 mines. However, since the within year production data is not available it is not clear how one can assign the production losses to floods, even if we were able to use satellite remote sensing to identify which mines were inundated to what degree. - Regarding how "geographical variability" or "discrepancy" is associated with different ranking, we were referring to the V versus CV measure rather than the S versus R. Thank you for pointing this out, it will be made clearer.

———————————————————

**Histogram of change**

[Figure]

**Fig. 1.**

---

## Author Response (AR1)

**Consolidated response to comments**

**Comments from Dr. Money**

We agree with the comment pointing out the limitations of our approach: "it would have been interesting to see some empirical longitudinal dataset that considers e.g. changes in production/ output at specific mining sites, correlated to observed rainfall, with appropriate adjustments. It would help substantiate the argument that extreme rainfall has material consequences for mining outputs, with the attendant financial consequences this represents. As it stands, that assumption seems intuitively correct, but it is hard to calibrate rainfall with other effects on mining output - which is necessary to calibrate the significance of this work [...]. The authors are however obliged to make a number of assumptions in terms of the likely financial consequences associated with 'excess exceedance' of 1 day/ 100 year and 30 day/ 10 year rainfall. These look somewhat generic, in particular assumptions of 'disruption in production' of 12.5% and 'destruction of NAV' of 10% (qualified as "likely a low number"). Applying these against the companies' revenue and asset value numbers generates a significant result - but this aspect of the methodology seems rather unsubstantiated: certainly when compared to the care given to the rainfall modelling aspect."

We considered using data from annual reports of mining companies for a longitudinal analysis. Some challenges we faced in developing such an analysis were:

1) Asset level impact data is not reported, and hence a direct quantification of the impact is not readily available
2) For significant losses, mining company annual reports may contain such information. However, we have not found a way to easily digitize this information. Specific instances of loss can indeed be identified. We could look at production losses (see an example below), but attributing them in each case to weather events may still be tedious since labor disputes or other market driven factors may be responsible and their effect may or may not be readily available. However, we are trying presently to gather more data and relate such events to losses through a Bayesian framework, for a later paper.
3) The return period associated with each event that may be reported is likely to vary substantially. If we had a lot of such events for which we had data, and were able to estimate the return period for each, we could indeed develop a loss probability distribution associated with extreme rainfall events. However, noting that multiple events at multiple mining sites may happen in a given year, as we demonstrate in the paper, it is not clear whether we can disentangle this information from the mining company reports and the rainfall data.

Some precisions regarding those issues were added in the discussion of pages 30-31.

Given these challenges, we took a forward (as opposed to inverse) modeling approach and sought to create an index where a standardized exposure could be identified from a climate perspective, noting that mine facilities are indeed designed to address extreme rainfall or drought events with a standard that relates to a nominal return period or probability of exceedance. This threshold may indeed vary by country and company, and by the amount and actual period of rainfall data used for the analysis.  This complicates the analysis as well. However, the principle we advanced was that we could indeed identify from long records what the thresholds should be at each mine, and hence what the exposure

characteristics with respect to those thresholds would be. If the mining company were asked to then disclose their design thresholds, and potential financial exposure if the threshold were exceeded, then a modified version of the analysis we present that used weights at each site that reflected the site by site exposure could be readily developed. The web app we have created would allow for such an analysis and also for its uncertainty analysis. It could be used internally by the mining company to assess and calibrate their potential exposure against loss data that may be available to them, or estimates, and then the result would be reported as part of their financial disclosure.

We felt that getting the idea out in the academic literature was a first step towards moving in this direction, and we have tried to add a discussion along the above lines to the revised version of the paper, thanks to your comment.

We can for instance investigate given events such as the Queensland floods. As shown in the plot below, commodity production of nearly all mines dropped in 2011.

[Figure]

Even for this example, where we know from news media that production was halted due to flooding for 6 months or longer for most mines, attribution to such events in a longitudinal analysis may not be easy. For the year in question, one could take the % drop in production relative to the prior year, or the

average of the prior and the succeeding year. However, there are other years with significant production drops across several of the mines where floods may or may not be an issue, and just for this region, one would have to build a statistical model with appropriate covariates to isolate the flood effects in a longitudinal sense. This is of interest, and we do intend to consider developing such a methodology to support risk analysis for the mining industry

As stated on point 2) above, we are planning to tackle the topics listed above in a more thorough paper on decision-making, which requires further data gathering.

With regards to your comment on the financial consequences of "excess exceedance," we acknowledge that the financial aspects of our approach are uncalibrated or in your words unsubstantiated at this point. Our primary assumption here is that for a particular rarity of the extreme event, relative to the design levels used by the mine, one could potentially expect a certain fraction of the NAV to be exposed. What this fraction is, and what an appropriate threshold is for such an impact are tuning parameters that indeed have not been worked out yet. Consequently, in this paper we focused on the spatial correlation structure of the exposure in the tails of the rainfall distribution, since we saw early on that this sort of clustering was emerging as a concern even for extreme rainfall events. Using the proportional loss idea basically only allows us to make a relative comparison of the exposure of the companies, and the numbers we cite in that regard as to the fraction of NAV exposed are in that spirit. We have tried to clarify this further in the revised version.

This is an aspect we hope to tackle in future research, which will necessitate extensive data gathering. Part of our goal is to encourage disclosure by having companies challenging the aspects of the paper that rely on simplified (albeit logical) estimations of financial loss once it is published. Our hope is that this will encourage better disclosure to allow investors, companies and regulators to better understand the risks which impact mining stakeholders.

**Comments from Dr. Wang**

- Regarding the issue of presenting devising risk index in part 2 from more hydrological perspectives, it is something that we have added to a little bit. We are currently in the process of gathering data on failed tailings dams to attempt to build our understanding of this issue. Overall, it has been a challenging process, we have been limited by data access in that regard, and this and the fact that overtopping is one of the major reason behind dam failures led us to consider precipitation data. However, we definitely hope to include more hydrological considerations in a later paper. We added sentences on pages 6, 7 and 8.
- Regarding the return level of the Queensland floods, this changes a lot from station to station. For instance, using the high quality data from the Australian Bureau of Meteorology (BMO, 2016): http://www.bom.gov.au/climate/change/hqsites/, the highest 30-day cumulative rainfall of the December 2010-February 2011 period corresponds to an approximately 8.5 year return-level in Barcaldine, a 3-year return level in Macknade and almost a yearly occurrence in Winton.
- The production loss assumptions were once again based on discussion we have had with experts rather than empirical analysis due to the lack of data access. For the Queensland floods, we noticed the distribution in the attached figure regarding the percent change in production between 2010 and 2011, based on data reported from 38 mines. However, since the within year production data is not available

it is not clear how one can assign the production losses to floods, even if we were able to use satellite remote sensing to identify which mines were inundated to what degree. However, by pulling together more such data, we hope to build a Bayesian framework to estimate the $C(p, d)$ per site.

**Histogram of change**

[Figure]

- Regarding how "geographical variability" or "discrepancy" is associated with different ranking, we were referring to the V versus CV measure rather than the S versus R. Thank you for pointing this out, this was addressed on page 28 (line 5).
- Typos and figure quality were dealt with.

[revised manuscript text omitted]

**Commented [I1]:** This precision is in response to comment 2 by Dr. Wang and the suggestion to expand the discussion of devising risk index
in part 2 from more hydrological perspectives

[revised manuscript text omitted]